# Spatial patterning of chloroplasts and stomata in developing cacao leaves
Insuck Baek[1], Seunghyun Lim[2], Visna Weerarathne[3], Dongho Lee[4], Jacob Botkin [5], Silvas Kirubakaran[3], Sunchung Park [2], Moon S. Kim[1], Lyndel W. Meinhardt [2] & Ezekiel Ahn [2] ✉

Leaf development and the coordinated formation of its key components is a fundamental process driving plant growth and adaptation. In tropical species like cacao, flush growth, a period of rapid leaf expansion, is particularly dependent on the optimized spatial patterns of chloroplasts and stomata. In this study, we investigated the patterns in cacao leaves during growth Stage C, a phase marked by rapid chlorophyll accumulation. Microscopic image data revealed significant acropetal variations in the size and density of chloroplast clusters and stomata, with the largest values found near the leaf base, mirroring the leaf greenness gradient. These findings suggest a coordinated developmental sequence between chloroplasts, stomata, and leaf ontogeny. A Support Vector Machine (SVM) model successfully classified distinct leaf regions based on these morphological features (>80% accuracy), highlighting the potential of machine learning applications in this area. Our results provide novel insights into the spatial coordination of chloroplast and stomatal development during cacao leaf maturation, offering a foundation for future research on flush growth optimization. To the best of our knowledge, this is the first report that combines microscopic data and machine learning analysis to investigate the leaf developmental process at stage C in cacao.

Cacao (*Theobroma cacao* L.), a multi-purpose tree species, is integral to the multi-billion-dollar chocolate industry worldwide and supports the economy and livelihoods of millions of smallholder farmers in tropical regions[1,2]. Thriving in shaded environments, cacao trees naturally grow as understory plants in forestry/agroforestry settings. Cacao undergoes alternating periods of growth and quiescence, marked by a distinctive flush cycle[3]. During flush periods, numerous leaves are rapidly produced and expand in optimal growth condition[4,5]. This period of rapid leaf development is particularly sensitive to environmental conditions. Daymond and Hadley demonstrated that environmental conditions such as temperature and light integral significantly influence early vegetative growth, chlorophyll fluorescence, and chlorophyll content in cacao with different genotypes exhibiting varying responses[6]. It is known that cacao leaves undergo a distinctive color transition during this flush cycle, progressing from red or pink (Stage A) to pale brown (Stages B & C), then light green (Stage D) as chlorophyll rapidly accumulates, and finally reaching full maturity with a dark green color (Stage E). This color progression reflects significant underlying physiological and biochemical changes. The most rapid chlorophyll accumulation occurs at Stages C and D, coinciding with leaf expansion[7]. Understanding the physiological and developmental changes during this crucial flush growth phase, especially at Stage C, is essential for optimizing cacao productivity. As the ancestral members of the plastid family, chloroplasts are not only responsible for photosynthesis but also play crucial roles in various metabolic and signaling pathways essential for plant growth and development[8]. The dynamic nature of plastids, which can interconvert between different forms like chloroplasts, amyloplasts, and chromoplasts in response to developmental and environmental cues, further underscores the importance of studying their development, particularly during the flush cycle in cacao[8]. In cacao, the transition from proplastids to functional chloroplasts during leaf maturation, particularly during the rapid chlorophyll accumulation phase (Stage C), is likely accompanied by significant changes in plastid morphology and spatial organization. Importantly, these changes are likely tightly

[1]Environmental Microbial and Food Safety Laboratory, Agricultural Research Service, United States Department of Agriculture, Beltsville, MD, 20705, USA. [2]Sustainable Perennial Crops Laboratory, Agricultural Research Service, United States Department of Agriculture, Beltsville, MD, 20705, USA. [3]Grape Genetics Research Unit, Agricultural Research Service, United States Department of Agriculture, Geneva, NY, 14456, USA. [4]Soybean Genomics & Improvement Laboratory, Agricultural Research Service, United States Department of Agriculture, Beltsville, MD, 20705, USA. [5]Department of Plant Pathology, University of Minnesota, St. Paul, MN, 55108, USA. ✉e-mail: ezekiel.ahn@usda.gov

coordinated, in both space and time, with stomatal development at specific locations within the leaf to ensure the proper balance between light harvesting, carbon fixation, and water use.

Chloroplast membrane development and chlorophyll synthesis have been reported to occur concurrently[9]. In cacao leaves, chlorophyll is synthesized slowly during the initial phases of leaf expansion, with significant synthesis occurring only after full leaf expansion is achieved[4]. Consequently, young leaves lack greenness during the early stages of flush growth. This absence of green color is also attributed to the initial small size and sparse quantity of chloroplasts[10]. Previous studies have also found correlations between chloroplast development, leaf ontogeny, and phenology, particularly in other plant species[3,4,11]. However, spatially and systematically generated data on leaf chloroplast or chlorophyll synthesis in cacao leaves are scarce. This limits our understanding of temporal leaf developmental functions and modeling of leaf carbon metabolism during flush growth. Furthermore, the specific spatial relationships between chloroplasts and stomata during this critical developmental window remain largely unexplored in cacao leaves.

Stomatal developmental patterns in cacao leaves were found to correlate with chlorophyll density, cuticle thickness, and leaf age[3]. Research in other species has shown that stomatal characteristics can vary considerably both within individual leaves and across larger leaf areas, highlighting the heterogeneity of stomatal development[12]. In young leaves, functional stomata initially appear on the midrib and mature veins, with their development extending into the interveinal regions as the leaf expands[3]. The delayed functionality of these later-developing stomata, which only become active in the final stage of leaf expansion, is likely an adaptive strategy to reduce transpirational water loss during the crucial flush growth phase[13]. This close association between stomatal development and leaf growth is further evidenced by the typical location of fully functional stomata on the green veins of etiolated leaves[3,13].

Despite the dynamic patterns of chloroplasts and stomata during cacao leaf development, a comprehensive understanding of their systematic development and morphogenesis analyses across distinct leaf growth phases remains elusive. Notably, the detailed spatial coordination between chloroplast and stomatal development during the critical Stage C phase of cacao leaf growth has not been thoroughly investigated. The complexity and spatial heterogeneity of these developmental patterns necessitate advanced analytical tools capable of deciphering intricate relationships within large datasets. Machine learning, with its capacity to identify patterns and make predictions from complex data, offers a promising avenue for unraveling the dynamics of stomatal and chloroplast development in cacao leaves.

Recent technological advancements, particularly in imaging and computational power, have facilitated the analysis of extensive and complex datasets in plant genomics and phenomics[14]. Concurrently, machine learning has emerged as a pivotal tool for extracting meaningful insights from complex data[14]. Supervised machine learning has driven rapid progress in diverse biological domains, including plant science[15]. These computational approaches hold immense potential for unraveling the complexities of stomatal and photosynthetic machinery. For instance, Xie et al.[16] demonstrated the effectiveness of machine learning in analyzing complex stomatal patterning traits in maize, enabling rapid phenotyping and subsequent QTL mapping. Li et al.[17] combined machine learning and remote sensing to model seasonal variations in the slope of stomatal conductance to photosynthesis for C3 and C4 crops. Costa et al.[18] developed a machine learning-based method to accurately and rapidly quantify stomatal density, size, and aperture in citrus leaves. Building upon these foundational studies, this study leveraged the capabilities of 'Support Vector Machine (SVM),' a well-established machine learning algorithm, to assess both linear and non-linear classification tasks in our leaf subsamples. SVM's ability to handle high-dimensional data and identify non-linear relationships makes it particularly well-suited for classifying and characterizing the intricate patterns of chloroplast and stomatal morphology. For non-linear problems, SVM was employed through a kernel function, which allowed the algorithm to map input data into a higher-dimensional space where a linear hyperplane could effectively separate the classes of interest.

To address key knowledge gaps in the phenological, physiological, and ontogenic aspects of chloroplast and stomatal development during cacao flush leaf growth, we investigated the spatial dynamics of chlorophyll, stomatal morphology, and venation patterns during the peak chlorophyll accumulation phase (Stage C) in developing leaves of cacao genotype Pound 7. This study represents a machine learning based approach to understanding cacao leaf development by focusing on the detailed spatial relationships between chloroplasts and stomata during this critical phase, providing a valuable starting point for investigating cacao leaf developmental processes. To assess spatial variations along with leaf expansion and maturation, we segmented the abaxial sides of leaves along the lateral veins into approximately 25–30 segments per leaf (Fig. 1a). These segments served as regions of interest for microscopic imaging of the spatial arrangements of chloroplasts and stomata (Fig. 1b). Image analysis characterized essential morphological traits of both entities at macro- and microscopic scales, covering 7652 chloroplast clusters and 11,809 stomata. Chloroplast clusters were defined as groups of individual chloroplasts that were in close proximity to each other within the imaged area, potentially originating from multiple cells. This enabled a descriptive assessment of the parameters mentioned above. This data was further analyzed using an SVM approach to systematically examine the complex interplay between these parameters, enriching our understanding of cacao flush leaf developmental processes.

## Results

### Chloroplast variation patterns in Pound7 cacao leaves

The size of chloroplast clusters in the interveinal regions was analyzed using seven morphological traits: area, length, width, length-to-width ratio (LWR), perimeter, circularity, and the distance between the intersection of length and width and the center of gravity (IS and CG). This analysis was performed at two levels: 1) within individual leaf segments defined by hierarchical vein sequences and 2) by categorizing these segments into four groups (G1–G4) based on their position along the leaf's vertical axis (Fig. 1a). All seven traits showed significant spatial variations ($p < 0.0001$) across individual leaves and combined groups of three Stage C Pound 7 leaves (Fig. 2a–d). Tukey's HSD test indicated a significant increase in the area size of chloroplast clusters near the leaf base (G4) and the interveinal segments between first-order veins compared to other leaf segments. Similar patterns were observed for the other six morphological traits, indicating significant correlations with the size dimensions of chloroplast clusters in each interveinal leaf segment (Supplementary Fig. S1). Supplementary Fig. S2 provides a detailed visualization of the mean and 95% confidence intervals for these chloroplast cluster morphology traits across individual leaf positions and grouped regions. A two-sample $t$-test revealed left-right asymmetry in the dimensions of chloroplast clusters between the left and right segments of each cacao leaf tested ($p < 0.0001$). For instance, the average area size of the chloroplast clusters was larger on the right side ($36,833.9 \pm 374.62\ \mu m^2$, $n = 3786$) than on the left side ($29,873.7 \pm 370.73\ \mu m^2$, $n = 3866$) (Supplementary Fig. S3).

### Stomata area size, shape, and variation patterns

Stomatal area varied across individual leaf segments, interveinal groups (G1–G4), and vein positions ($p < 0.0001$) (Fig. 3). As observed with chloroplast clusters, the largest stomata were predominantly located near the leaf base in most leaves (Fig. 3a, c, d), with the exception of Leaf 2 (Fig. 3b). This pattern was also reflected in stomatal perimeter, length, and width (Supplementary Fig. S2b, c, d). Left-right asymmetry in stomatal area was observed; larger stomata were mainly found on the right side of two leaves (Fig. 3a, c), while Leaf 2 showed the opposite trend (Fig. 3b). However, when data from all three leaves were combined, no significant difference in stomatal area was found between the left and right sides. Supplementary Fig. S4 provides a detailed visualization of the mean and 95% confidence intervals for these stomatal morphology traits across individual leaf positions and grouped regions.

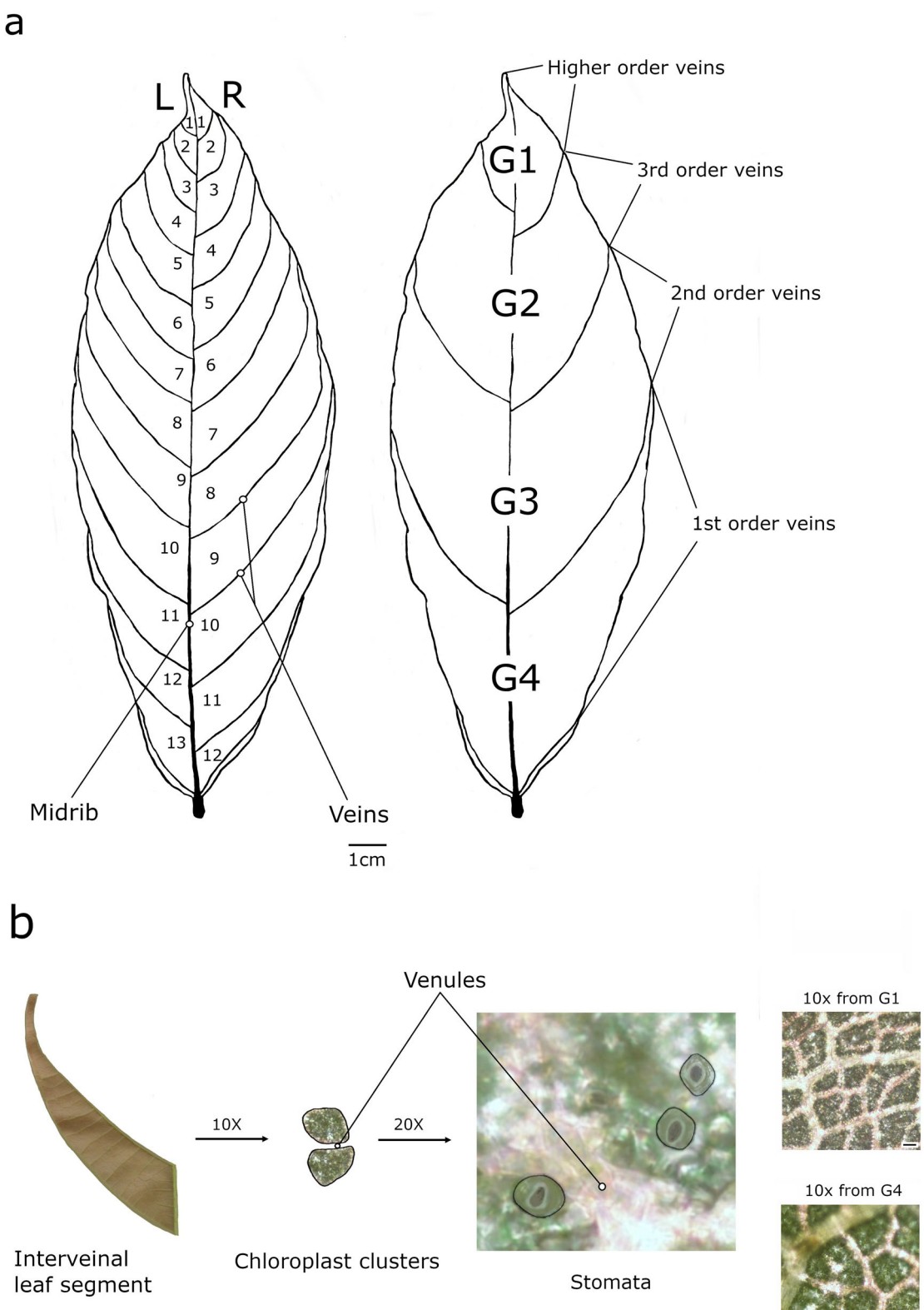

**Fig. 1 | Leaf interveinal segmentation and microscopic visualization of chloroplast clusters and stomata. a** Interveinal segmentation from top to bottom on each left (L) or the right (R) half of the abaxial side of a representative cacao leaf (genotype Pound7). These segments were further grouped into four zones (G1–G4) from the leaf apex to the base to facilitate visualization and analysis. **b** Each leaf segment was imaged under a microscope at 10X magnification to visualize the variation of chloroplast clusters in the interveinal regions. The lamina between venules within each segment was then imaged at 20X magnification to visualize stomatal variation and morphology. Outlines of guard cells are provided for illustrative purposes. Representative images of chloroplast clusters for G1 and G4 zones are also shown at 10X magnification to illustrate differences in size and to provide examples of the clusters analyzed in this study (scale bar = 100 μm).

**Fig. 2 | Spatial variation of chloroplast clusters.**
**a–c** Heatmaps visualizing the variation in chloroplast cluster sizes across interveinal segments within leaf groups G1–G4. Darker green indicates larger average chloroplast cluster size. **d** Combined data from all leaves in groups. Capital letters (A, B, C) denote significant differences in chloroplast cluster size frequency between segments (Tukey's HSD test, $p < 0.0001$). Scale bars = 1 cm. Sample sizes: **a** 2339, **b** 3978, **c** 1335, **d** 7652.

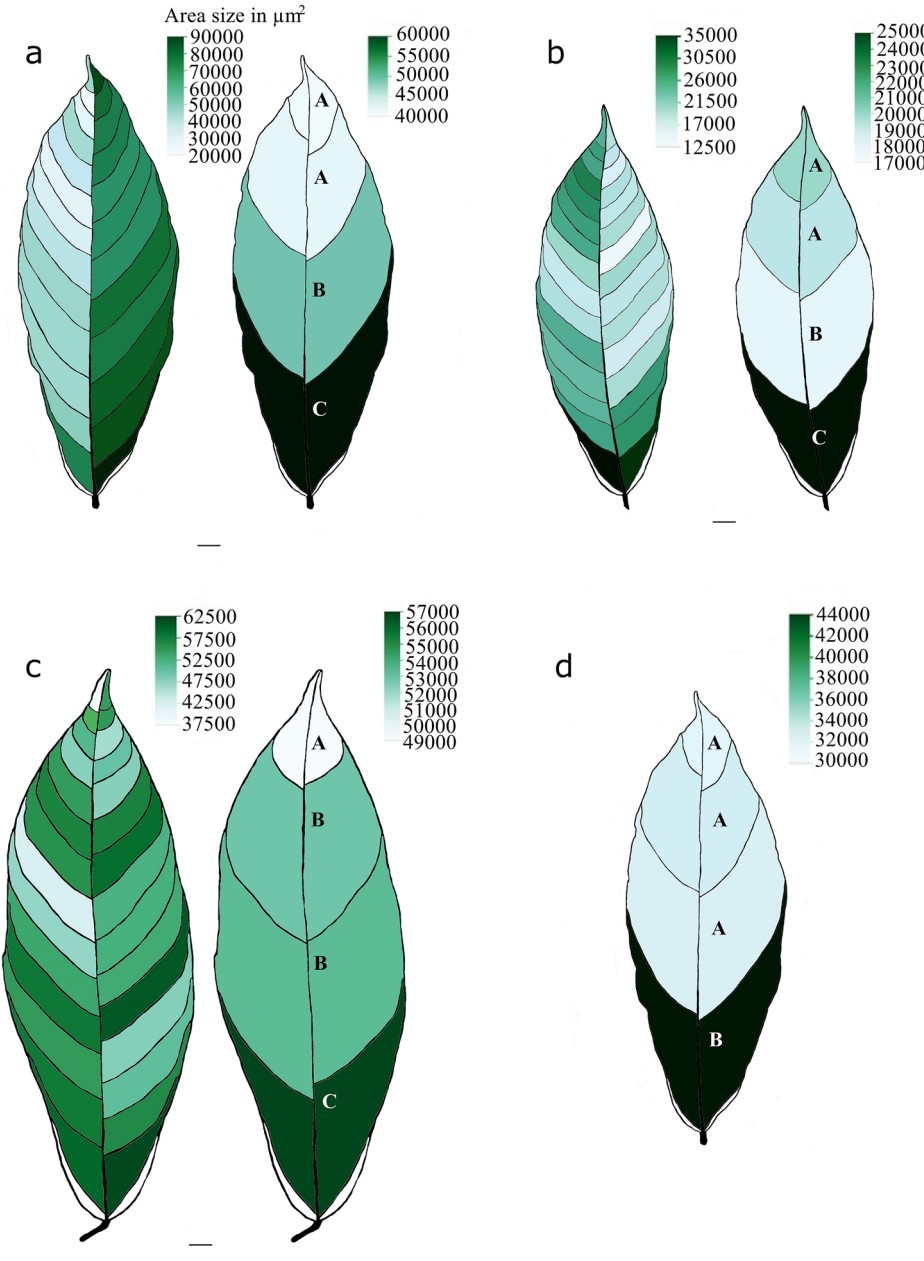

## Spatial variation in circularity of localized chloroplast groups and stomata

Significant spatial variations in the circularity of chloroplast clusters and stomata were observed across the G1–G4 groups (Fig. 4; $p < 0.05$). Other morphological traits, including LWR and IS and CG, also showed significant spatial variation across individual leaf segments and G1–G4 groups ($p < 0.05$). Although these differences were subtle, they consistently reached statistical significance. Notably, both chloroplast clusters and stomata tended to be more circular towards the leaf apex, indicating a correlation between the circularity of these two features ($r = 0.3$, $p < 0.006$).

## Correlations between chloroplast, stomata, and leaf segment morphological parameters

The relationships among morphological parameters of chloroplast clusters, stomata, and leaf segments are summarized in a correlation matrix (Fig. 5). The average size of chloroplast groups and stomata were positively correlated ($r = 0.65$, $p < 0.0001$). Leaf area also showed positive correlations with both the average area of chloroplast groups ($r = 0.28$,

$p < 0.0001$) and stomata ($r = 0.23$, $p = 0.04$). Leaf greenness was initially quantified by the ratio of green pixel area to each leaf segment area. However, as ImageJ represents higher intensity with lower RGB values (closer to 0), we reversed the scale (0–255) for the analysis presented in Fig. 5. This revealed a strong positive correlation between leaf greenness and the average area of both chloroplast groups ($r = 0.54$, $p < 0.0001$) and stomata ($r = 0.41$, $p < 0.0001$).

Blue color intensity demonstrated moderate positive correlations between the average area of chloroplast clusters ($r = 0.39$, $p = 0.0002$) and stomata ($r = 0.29$, $p < 0.0001$). Furthermore, circularity and IS and CG of both chloroplast groups and stomata showed significant correlations at $p < 0.05$ (Supplementary Data 1).

Hierarchical clustering analysis (Fig. 6) reveals distinct groupings among leaf traits. Macroscopic leaf segment traits, chloroplast traits, and stomatal traits predominantly cluster within their respective categories, underscoring their inherent structural and functional relationships. Notably, chloroplast and stomatal circularity cluster together, while leaf segment circularity forms a distinct cluster.

**Fig. 3 | Spatial variation of different-sized stomata in Stage C cacao leaves. a–c** Heatmaps illustrating the spatial variation of different stomatal area size categories (quantified based on the average stomatal area size, including guard cells) across interveinal leaf segments and the four-leaf groups (G1–G4) from top to bottom. Dark green segments represent areas with the largest stomata compared to others. **d** Combined data from all leaves. Capital letters (A, B, C) denote significant differences in stomatal area size frequency between combined leaf segments (Tukey's HSD test, *p* < 0.0001). Scale bars = 1 cm. Sample sizes: **a** 3048, **b** 5756, **c** 3005, **d** 11,809.

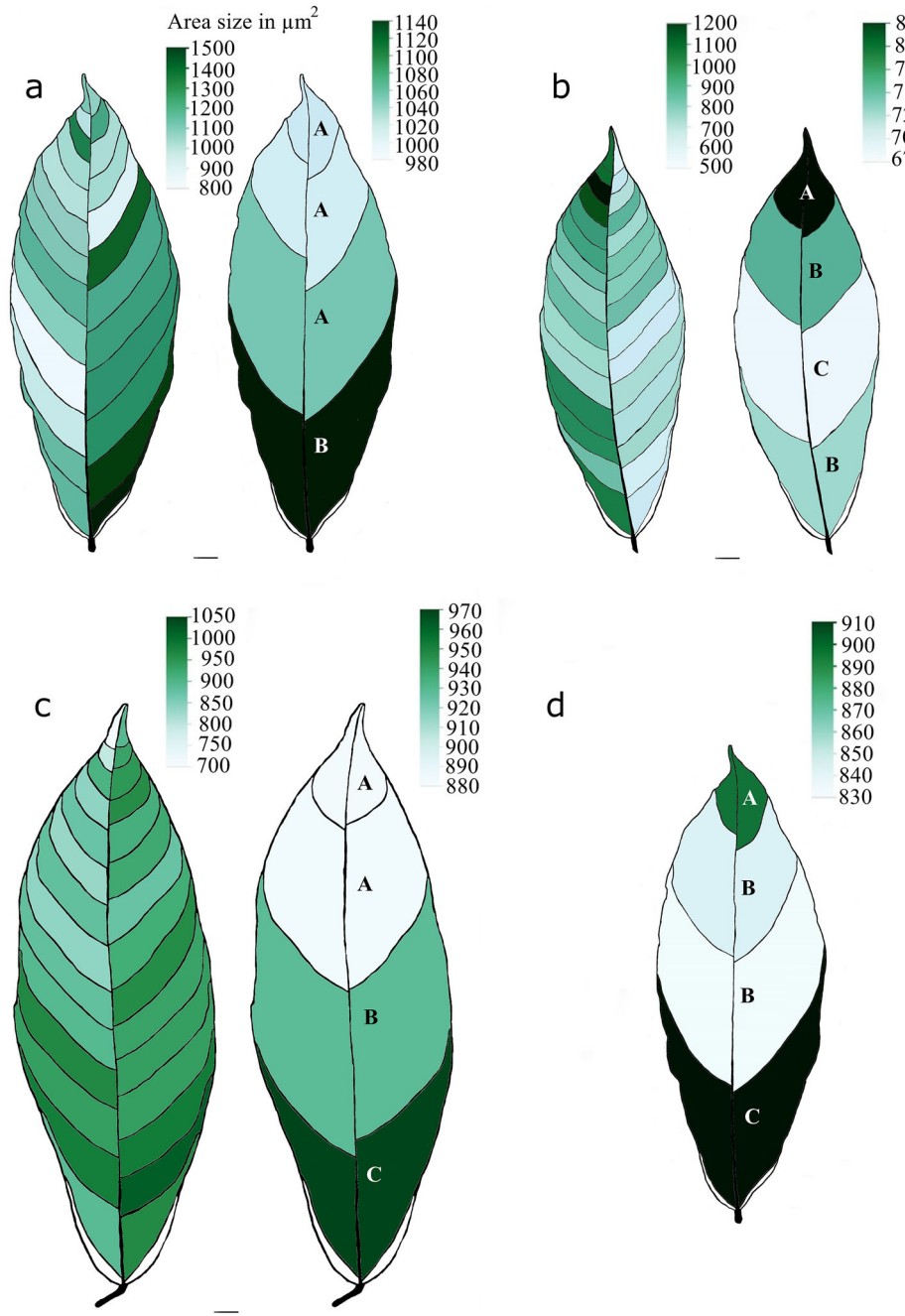

## SVM-based classification of chloroplast and stomata variation patterns

This study employed SVM with a radial basis function kernel (SVM-RBF) to classify patterns in the spatial variation of chloroplasts and stomata across leaf segments, as Hearst et al.[19] and Suthaharan[20] described. Two distinct SVM models were developed and trained on 80% of the manually measured data, while the remaining 20% was used for testing. This manually measured data was obtained using the SmartGrain software as a tool to assist in the tracing and quantification of chloroplast clusters and stomata. Samples were categorized into groups A and B (leaf groups G1-3 together and G4, respectively).

Figure 7 illustrates the relative importance of each morphological feature for classifying the spatial variation of chloroplasts and stomata across leaf interveinal segments, as determined by ANOVA *p*-value. Length, area, and LWR were identified as the most influential features for

classifying stomatal variation, while circularity, length, and width were pivotal for classifying chloroplast variation. Both microscopic visualization and SVM-RBF classification revealed distinct patterns in the variation of chloroplasts and stomata between basal and non-basal leaf regions. Table 1 displays SVM-RBF performance data based on all seven morphological features or the top three features identified above. Notably, the accuracy level reached 81.27% when all seven features were employed together for classifying stomatal variation, slightly higher than the 77.69% accuracy obtained using only the top three features. These results underscore that the top three features selected by ANOVA *p*-value for each model were sufficiently robust for accurate classification and interpretation of chloroplast and stomatal variation patterns across cacao leaf interveinal segments. Detailed performance metrics developed for two different sets of morphological features are provided in Supplementary Figs. S5 and S6.

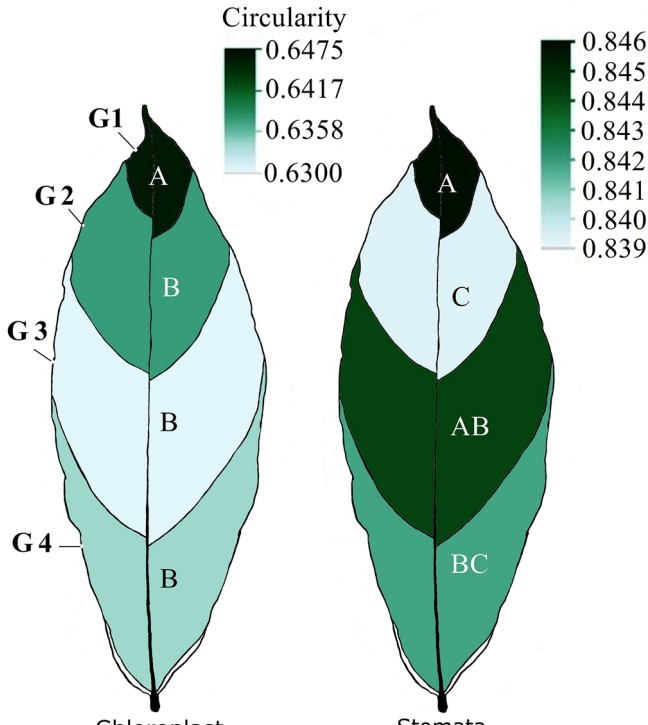

**Fig. 4 | Variation of circular chloroplast groups and stomata across G1- G4.** Different letters indicate significant statistical differences among G1, G2, G3, and G4 segments for the circularity of chloroplast groups and stomata (sample sizes: 7652 and 11,809, respectively).

## Discussion

This study reveals strong correlations between chloroplast cluster features, spatial patterns, stomatal morphology parameters, and leaf greenness, particularly within Stage C cacao leaves. These findings highlight the dynamic developmental processes occurring during this crucial phase of leaf maturation. Figure 8 proposes a hypothetical developmental sequence to explain the observed spatial and temporal variations in chloroplast clusters and stomata, offering insights into the progression from Stage C towards Stage D. The gradient of leaf greenness, increasing from base to apex, suggests an acropetal development process influenced by chlorophyll accumulation and venation patterns. This process likely reflects assimilated partitioning for both chlorophyll biosynthesis and stomatal development, laying the foundation for the physiological functions that will become fully established in Stage D.

In the developing Stage C cacao leaves, we interpreted larger stomata, especially those in the basal regions, as being further along in their developmental trajectory towards full functionality compared to the smaller stomata found in the apical regions. Furthermore, the analysis of chloroplast clusters, rather than individual chloroplasts, provides a unique perspective on the spatial organization of photosynthetic machinery within the developing leaf. This spatial organization may play a role in optimizing the sequential nature of photosynthesis, which, as Ruban emphasizes, involves a cascade of energy transformation events occurring across vastly different timescales[21]. Photosynthesis initiates with photon absorption occurring at the femtosecond scale, followed by electron transfer at the micro- to milli-second scale, and culminating in carbon fixation and the accumulation of starch over minutes to hours[21]. The arrangement of chloroplasts into clusters could potentially influence the efficiency of these processes, for example, by affecting the coupling between light harvesting, electron transport, and carbon fixation[21].

Our findings on spatial heterogeneity of chloroplast cluster and stomatal area size and density in cacao leaves are consistent with research highlighting the variability of stomatal characteristics at both the individual pore level and across larger leaf areas[12]. While individual chloroplast morphology is important, cluster-level analysis captures the heterogeneity of chloroplast variation and its potential relationship to overall photosynthetic capacity. The size, shape, and density of chloroplast clusters likely reflect the developmental stage of the mesophyll tissue and its progression towards full photosynthetic competence. The observed variations in chloroplast clustering patterns may reflect the remarkable diversity and adaptability of photosynthetic antennae. As Ruban discusses, the light-harvesting antenna has been 'reinvented' multiple times during evolution, resulting in a wide variety of pigment-protein complexes[21]. This evolutionary history suggests a strong selective pressure to optimize light harvesting under diverse environmental conditions. For instance, larger, more densely packed clusters may indicate regions of the leaf with higher light-harvesting capacity, potentially due to increased chlorophyll density or optimized thylakoid organization within the chloroplasts[22]. Additionally, the wavelike spatial variation trends in chloroplast clusters and stomatal size observed in Supplementary Figs. S2a and S4a suggest a possible link to dynamic circadian growth processes, which regulate diurnal rhythms in plant development. However, our current study, focused on a single developmental stage (Stage C) under controlled conditions, did not assess circadian influences. Future research incorporating time-series imaging and gene expression analyses could elucidate whether these patterns are driven by circadian rhythms, enhancing our understanding of cacao leaf development.

Previous research highlighted stomatal responses to various micro- and macro-environmental factors, including water pressure deficit, light, humidity, temperature, and soil moisture[23]. Our findings suggest that the enlargement of stomata near the leaf base and first-order veins in Stage C cacao leaves may be an adaptation to reduce hydraulic resistance in these lower leaf regions. This would enable larger stomata to facilitate gas exchange without negatively impacting net assimilation or transport stored assimilates from source leaves to sinks. Conversely, the reduced stomatal area size near the leaf apex may be a strategy to minimize water loss, as depicted in Fig. 9. This water conservation strategy is likely associated with the increased transport distance and potential gravitational effects caused by the drooping tendency of the leaf apex. Such developmental adaptations underscore a possible trade-off between gas exchange and water conservation in young, actively growing cacao leaves. Notably, the observed stomatal variation pattern aligns closely with the spatial variation of chloroplast clusters, suggesting an interconnected developmental process. This is consistent with findings that stomatal development is tightly regulated by various signals that influence photosynthesis, water-use efficiency, and, ultimately, plant growth[24]. While our study reveals strong correlations between chlorophyll accumulation, stomatal morphology, and spatial growth patterns in Stage C cacao leaves, the causal relationships among these factors remain unclear. Future experimental approaches, such as genetic manipulation or detailed physiological assessments over time, could help elucidate whether chlorophyll accumulation drives stomatal development or vice versa and how these processes are coordinated during flush growth.

Previous research by Sack et al.[25] highlighted a correlation between midrib and first-order vein diameter and leaf size, while other veins showed no such relationship. Our findings further expand on this, revealing a clear correlation between venation patterns and the size and variation of stomata and chloroplasts across Stage C cacao leaves. This suggests distinct relationships between the circularity and size characteristics of chloroplast groups and stomata and the underlying venation patterns. These results contribute to a fundamental understanding of young cacao leaves' morphological and developmental processes. Furthermore, while chlorophylls are synthesized in chloroplasts and carotenoids in chromoplasts, originating from a common precursor, the genetic relationship between their biosynthetic pathways remains partially understood[26]. Variations in leuco-anthocyanin levels, which contribute to red pigmentation, correlate with total leaf tannin content and are influenced by factors like leaf age and cacao variety[27]. Additionally, chlorophyll content, especially Chl b, and stomatal conductance are critical determinants of net photosynthetic rate[28].

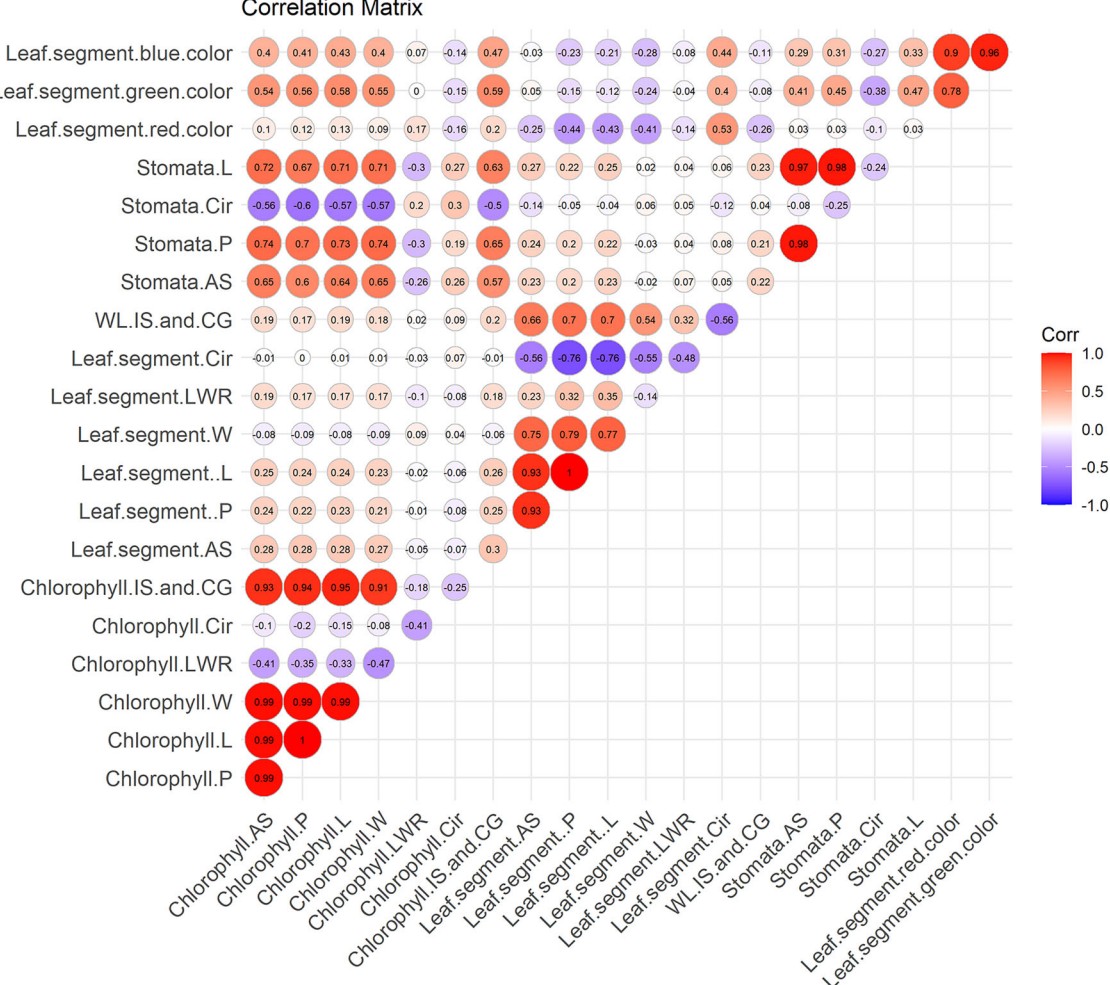

**Fig. 5 | Spatial correlations between leaf traits, chloroplast clusters, and stomata in Stage C cacao.** Leaf segment area, defined by midrib and secondary veins, correlates significantly with both chloroplast cluster area ($r = 0.28$, $p < 0.0001$) and stomatal area ($r = 0.23$, $p = 0.04$), along with other size-related traits. A strong positive correlation also exists between chloroplast cluster area and stomatal area ($r = 0.65$, $p < 0.0001$). Furthermore, leaf greenness (inversely proportional to green pixel intensity) is significantly correlated with both chloroplast cluster area ($r = 0.54$, $p < 0.0001$) and stomatal area ($r = 0.41$, $p < 0.0001$). Detailed correlation values and $p$-values are provided in Supplementary Data 1. AS area size, P perimeter, L length, W width, LWR length-to-width ratio, Cir circularity, and Corr correlation.

Previous research in soybean has demonstrated negative correlations between leaf size and shape but positive correlations between both of these traits and chlorophyll content[29]. Similarly, in our study of Stage C cacao leaves, we observed strong correlations between most morphological leaf traits, including those of chloroplast groups and stomata, and the leaf area of interveinal segments. Besides, our study further found interesting correlations between whole leaf circularity and chloroplast group size ($r > 0.99$, $p < 0.05$) (Supplementary Data 1 for the complete correlation matrix). These results suggest a potential link between leaf shape and chloroplast developmental patterns across the leaf, but given the limited sample size, further research with a larger dataset is warranted. Hierarchical clustering analysis revealed a unique morphological connection between the circularity of chloroplast groups and stomata, distinct from the circularity of leaf segments. This clustering suggests a potential functional or evolutionary link between these traits, possibly related to processes like photosynthesis or gas exchange, where the shape and arrangement of chloroplasts and stomata are crucial.

Machine learning has been increasingly applied to various aspects of cacao research, including disease detection[30], fermentation optimization[31], healthy plant and pod prediction[32], and wild cacao canopy parameterization[33]. However, its use in characterizing microscopic data, specifically on the morphology of photosynthetic machinery, has been limited. This study employed machine learning approach to elucidate key determinants of stomatal and chloroplast developmental features in Stage C cacao leaves, achieving over 75% accuracy in classifying entities of interest with only a limited number of predictors. The high classification accuracy observed for the basal leaf region (Group B) in both stomata and chloroplast variation models, as evidenced by the near-perfect scores in the test sets (Supplementary Figs. S5, S6), suggests that this region possesses distinct morphological characteristics that set it apart from other leaf regions. In contrast, the comparatively lower accuracy for the non-basal regions (Groups A/G1-G3) could be attributed to the inherent heterogeneity within these regions, encompassing a wider range of stomatal and chloroplast sizes and shapes. This observation aligns with the hypothesis of an acropetal developmental process, where the basal region, being more mature, exhibits greater uniformity in its cellular structures, while the non-basal regions, undergoing active development, display a greater degree of variability. The machine learning models' ability to effectively capture this distinction between basal and non-basal regions underscores the potential of such approaches for deciphering complex developmental patterns in microscopic data. This successful application in characterizing stomatal and chloroplast variation in cacao leaves lays a foundation for future investigations leveraging machine learning to understand the spatial and temporal variation of leaf anatomical structures across various genotypes and developmental stages.

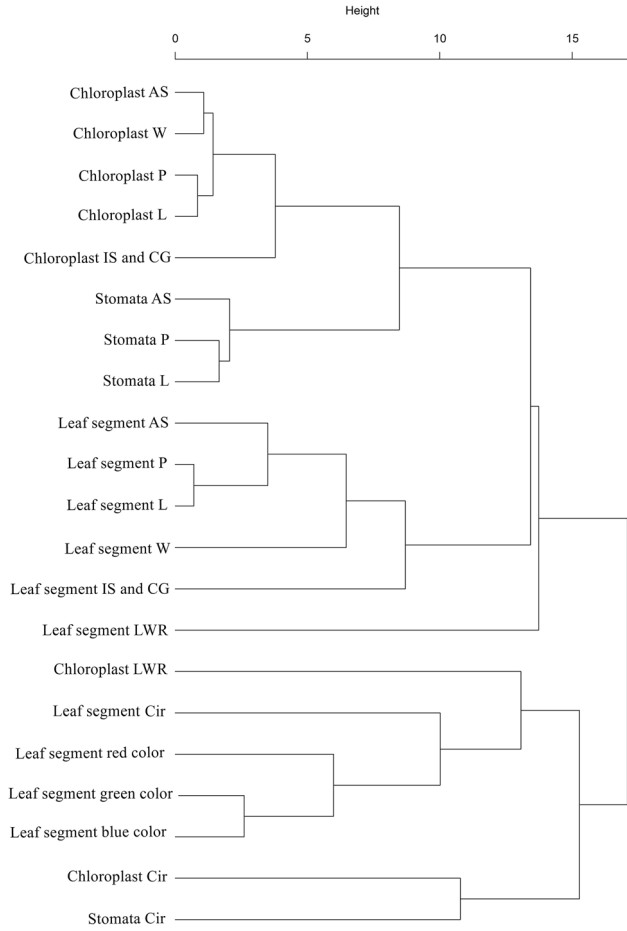

**Fig. 6 | Hierarchical clustering of leaf, chloroplast, and stomatal traits.** The dendrogram depicts hierarchical clustering of macroscopic leaf segment traits, microscopic chloroplast traits, and stomatal traits. AS area size, P perimeter, L length, W width, LWR length-to-width ratio, Cir circularity.

Our findings offer valuable insights into the developmental biology and basic physiology of cacao, a tropical crop species that has received relatively less attention compared to others. Beyond the morphogenesis of photosynthetic machinery, physiological factors, such as hormonal regulation, also play a significant role in the flush growth cycle and overall plant development. For example, abscisic acid (ABA) is a crucial driver of various plant processes, including stomatal movements and seasonal plant growth patterns linked to plant water dynamics[34,35]. Although our study examines the spatial patterns of chloroplasts and stomata, the role of phytohormone signaling in establishing these patterns remains unexplored. While the specific role of ABA in cacao leaf development requires further investigation, its known influence on stomatal regulation suggests it may contribute to the observed spatial patterns of stomata and chloroplasts. Ultimately, a deeper understanding of these complex interactions between leaf development, physiology, and environmental factors will inform strategies to optimize cacao growth and productivity in a changing climate.

## Materials and methods
### Plant materials
Cacao genotype Pound7 leaves at growth Stage C, exhibiting the characteristic color transition from brown to light green and indicating active chlorophyll accumulation, were collected from clonally propagated trees in a greenhouse at the USDA-ARS, Beltsville, MD, USA. Three leaves were randomly sampled from separate trees for three replicates. The greenhouse was maintained at day/night temperatures of 85 °F/75 °F, with ~60% relative humidity, a minimum 12-hour photoperiod, and a light intensity of 325 µmol/m²/s[36].

### Leaf scanning and microscopy
Collected leaves were immediately transported to the laboratory with their petioles immersed in sterile water-filled vials to maintain freshness. Whole leaves were scanned on adaxial (upper) and abaxial (lower) surfaces using a Canon Color ImageCLASS MF656Cdw scanner. The abaxial surface of each leaf was then divided into left and right segments along the lateral veins (Fig. 1a). The number of segments varied slightly among leaves (Leaf 1: L1-L13, R1-R12; Leaf 2: L1-L15, R1-R15; Leaf 3: L1-L14, R1-R15). Each leaf was further divided into four groups along its vertical axis (Fig. 1a). Leaf segments were carefully separated using a scalpel and imaged under a Nikon

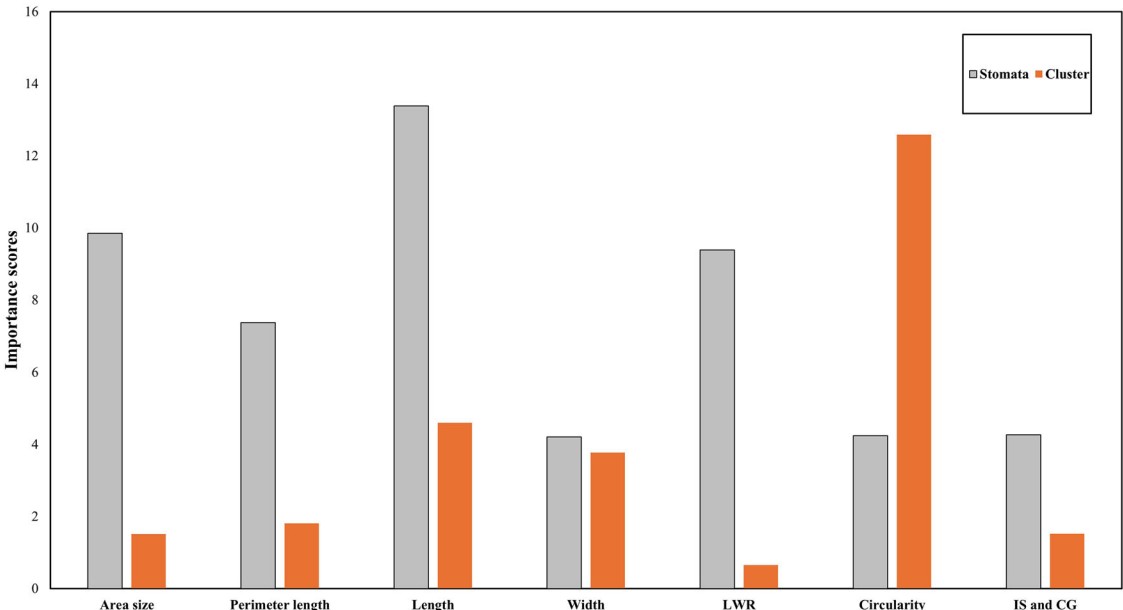

**Fig. 7 | Feature importance for classifying stomatal and chloroplast spatial variation.** The bar graph depicts the relative importance of seven morphological features in classifying the spatial variation of stomata (gray bars) and chloroplasts (orange bars) in Stage C cacao leaves. Higher bars indicate greater importance for classification. DPC displayed patterns of chloroplasts, DPS displayed patterns of stomata, LWR length-width ratio, IS and CG distance between the intersection of length and width (IS) and center of gravity (CG).

**Table. 1 | Model performance comparison against seven morphological features and the top three features used for stomata and chloroplast classifications**

|  | Train set | | | | Test set | | | |
|---|---|---|---|---|---|---|---|---|
|  | Accuracy (%) | F1 | Precision | recall | Accuracy (%) | F1 | Precision | recall |
| Stomata with full-features | 100 | 1 | 1 | 1 | 81.27 | 0.81 | 0.86 | 0.81 |
| Stomata with selected features | 96.71 | 0.97 | 0.97 | 0.97 | 77.69 | 0.77 | 0.82 | 0.78 |
| Chloroplast with full-features | 100 | 1 | 1 | 1 | 76.96 | 0.76 | 0.84 | 0.77 |
| Chloroplast with selected features | 100 | 1 | 1 | 1 | 76.96 | 0.76 | 0.84 | 0.77 |

This table compares the performance of the SVM-RBF models for classifying stomata and chloroplast variations using two feature sets: all seven morphological features and the top three features selected by ANOVA. Performance metrics (accuracy, F1-score, precision, and recall) are reported for both the training and test sets.

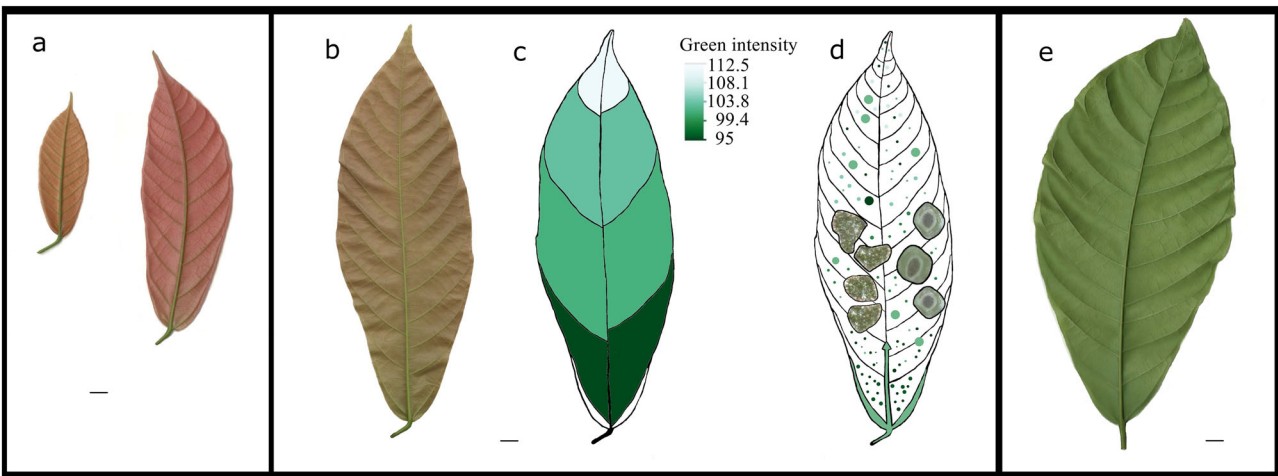

**Fig. 8 | Proposed model of chloroplast clusters, stomatal development, and leaf maturity in cacao. a** Scanned images of early-stage cacao leaves (Stages A and B). **b** A scanned image of a Stage C cacao leaf (leaf 3) was used in this study. **c** Heatmap depicting leaf greenness intensity, with darker shades representing higher chlorophyll content (green intensity: 0 = maximum, 255 = minimum). **d** Hypothetical illustration of the developmental sequence of chloroplast clusters and stomata in a Stage C cacao leaf, based on the spatial variation of their morphological features. The increasing green pigmentation towards the leaf base suggests an acropetal leaf development process, likely interconnected with vein development and the transport of resources through the midrib. **e** Scanned image of a Stage D leaf. Scale bars = 1 cm.

ECLIPSE E600 microscope equipped with a Nikon DS-Ri2 camera, with an image bit depth of 24-bit RGB color. For the purposes of this study, chloroplast clusters were defined as groups of individual chloroplasts that were in close proximity to each other within the imaged area, potentially originating from multiple cells. Chloroplast clusters within the interveinal leaf segments were analyzed at 10X, with clusters connected by more than 50% of their area considered a single unit. Stomatal morphology, including guard cells, was assessed at 20X. To minimize the influence of environmental factors, stomatal area size was measured instead of aperture. All images were saved in JPEG format for further analysis.

### Image analysis

The morphological traits of whole leaves, leaf segments, chloroplast clusters, and stomata were quantified using SmartGrain software (version 1.3)[37]. Although primarily designed for seed morphology analysis, SmartGrain was suitable for this study due to its capability to measure key traits such as area, length, width, length-width ratio (LWR), perimeter, circularity, and the distance between the intersection of length and width (IS) and the center of gravity (CG). However, automated detection and measurement of chloroplast clusters and stomata were not possible with the SmartGrain software. Therefore, these features were manually traced and measured using the software's tools. Overall, 7652 chloroplast clusters and 11,809 stomata were randomly selected and measured across the three leaves. Measurements for chloroplast clusters and stomata were recorded in micrometers (μm), while whole leaf and leaf segment sizes were measured in centimeters (cm). A complete list of all measured traits, their abbreviations, descriptions, and units is provided in Supplementary Data 2. Leaf greenness was assessed by measuring RGB values (0–255 scale, 0 = highest intensity) for whole leaves, leaf segments, and leaf groups using ImageJ 1.54 d software[38]. Images were opened in ImageJ, and the 'Split Channels' function was used to separate the RGB color channels. The freehand selection tool was then used to manually outline the leaf or leaf segment area, excluding any background area. This defined the Region of Interest (ROI). The 'Measure' function was then used to obtain the mean RGB values (0–255 scale, 0 = highest intensity) within the whole leaves. All raw morphological data are available in Supplementary Data 1.

### Statistical and bioinformatics analysis

To assess the impact of leaf position on microscopic traits of chloroplast clusters and stomata, ANOVA followed by Tukey's HSD tests were conducted using JMP Pro 17 (SAS Institute, Cary, NC, USA). To compare specific groups, t-tests were also conducted using JMP Pro 17.

We computed the Pearson correlation matrix for the selected traits to investigate the relationships between leaf traits, excluding pairs with missing data to ensure accurate estimates. The statistical significance of each correlation was assessed by calculating p-values. A correlation plot was then generated to visually represent the strength and direction of associations between traits, with significant correlations highlighted.

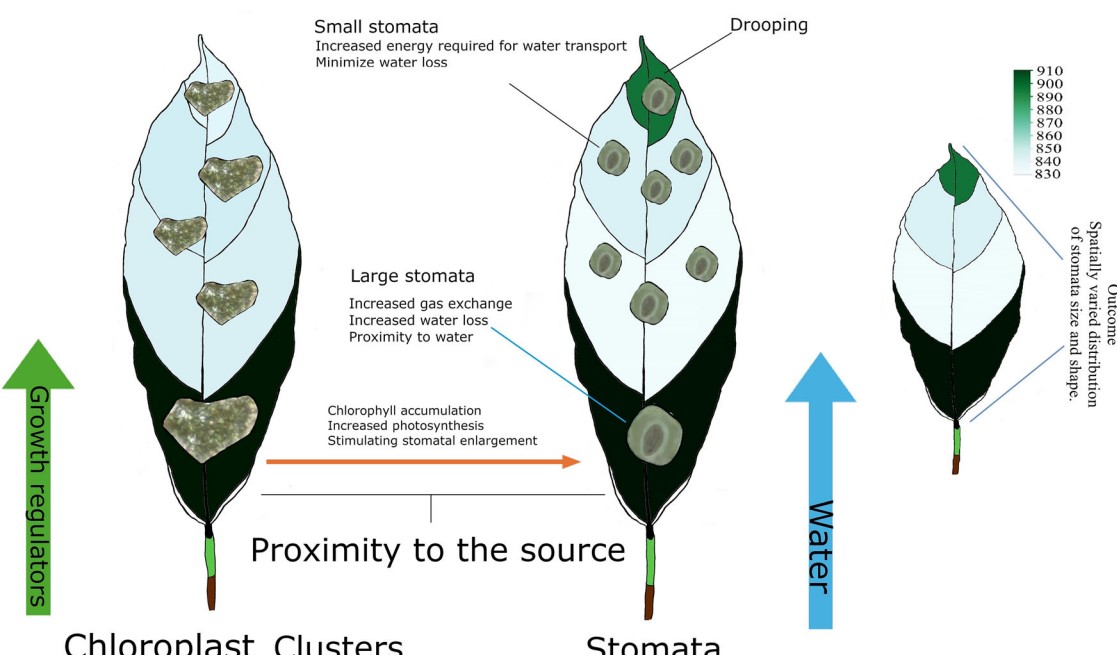

**Fig. 9 | Hypothetical model of stomatal area size, chloroplast variation, and resource transport in a Stage C cacao leaf.** The diagram illustrates the proposed relationships between the spatial variation of chloroplast groups and stomata size and the direction of water and assimilate transport within a developing cacao leaf. It highlights the potential influence of chlorophyll accumulation, water availability, and growth regulators on these patterns.

In addition to examining pairwise correlations, we employed hierarchical clustering analysis to better understand the relationships between leaf traits and stomatal characteristics. Data on chloroplast and stomatal traits from two datasets were aggregated by calculating the means of each trait within groups defined by leaf position and other relevant identifiers. The aggregated data sets were then merged based on a common identifier. The selected leaf traits were standardized using z-scores to ensure comparable distances between variables. This transformation converts the data to have a mean of zero and a standard deviation of one. Divisive hierarchical clustering (DIANA) was applied to the standardized data. DIANA starts with all data points in a single cluster and iteratively splits the clusters based on dissimilarity. The resulting dendrogram was visualized to assess the clustering patterns among the leaf traits.

**Support Vector Machine for prediction and validation**
SVM with a Radial Basis Function (RBF) kernel was used for non-linear analysis using MATLAB version 9.9 (R2019a)[19,20]. The RBF kernel can be described with an equation:

$$k(x_i, x_j) = exp\ exp(-\gamma \|x_i - x_j\|^2) \qquad (1)$$

The RBF kernel is defined by a parameter γ (gamma), which controls the influence of individual training points[39]. It is complemented by the C parameter, which balances the trade-off between maximizing the margin and minimizing classification errors. To determine the optimal values for γ and C, a grid-search method[40] combined with 10-fold cross-validation was employed. This approach helps find the best parameters and prevent overfitting, ensuring the model generalizes well to new data.

Two classification models were developed to identify stomata and chloroplast clusters. For the stomata model, the dataset consisted of 2506 samples, with each group containing 1253 samples. For the cluster model, the dataset included 2930 samples, with each group containing

1465 samples. Group A data was collected from leaf position groups G1, G2, and G3 on the leaf, while Group B data was collected from leaf position group G4 (basal part). The dataset was divided into training and testing sets, with 80% of the data used for training and 20% reserved for testing. Each sample was characterized by seven features: area size, perimeter length, length, width, LWR, circularity, and IS and CG.

**Statistics and reproducibility**
This study employed a hierarchical sampling design to ensure robust and reproducible results. Three biological replicates were used, each consisting of a single leaf collected from a separate, clonally propagated Pound 7 trees grown under controlled greenhouse conditions. Within each leaf, 25–30 interveinal segments were analyzed. From these segments, a total of 7652 chloroplast clusters and 11,809 stomata were randomly selected and manually measured across all three leaves using SmartGrain software (version 1.3)[37]. Statistical analyses were performed using JMP Pro 17 (SAS Institute, Cary, NC, USA)[41]. The impact of leaf position on the morphological traits of chloroplast clusters and stomata was assessed using ANOVA followed by Tukey's HSD post-hoc tests. Pearson correlation coefficients were calculated to investigate the relationships between leaf traits, chloroplast cluster traits, and stomatal traits. Hierarchical clustering analysis with the DIANA method explored the relationships among the various measured traits. Before clustering, the data were standardized using z-score.

**Reporting summary**
Further information on research design is available in the Nature Portfolio Reporting Summary linked to this article.

**Data availability**
Phenotypic data are available in Supplementary Data 1. All other data are available from the corresponding author (or other sources, as applicable) on reasonable request.

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

## Acknowledgements

We thank Dr. Zhuangji Wang (Adaptive Cropping Systems Laboratory, USDA-ARS) and Dr. Minhyeok Cha (Environmental Microbial and Food Safety Laboratory, USDA-ARS) for their valuable insights and contributions during manuscript preparation. We are also grateful to the reviewers for their constructive feedback.

## Author contributions

E.A. designed and directed the overall study. L.W.M., M.S.K., and E.A. obtained funding for its execution; I.B. performed machine learning analysis. S.L. and E.A. conducted bioinformatics and statistics. S.P. provided resources. S.P., M.S.K., L.W.M., and E.A. provided supervision. V.W., D.L., J.B., S.K. provided supports developing methodology. All authors contributed to the writing and editing of this manuscript.

## Competing interests

The authors declare no competing interests.
