## [Transparent Peer Review file · Communications Biology]

Spatial patterning of chloroplasts and stomata in developing cacao leaves

Corresponding Author: Dr Ezekiel Ahn

Version 0:

Reviewer comments:

Reviewer #1

(Remarks to the Author)

The article titled "Spatial Patterning of Chloroplasts and Stomata in Developing Cacao Leaves" by Insuck Baek et al. investigates the spatial distribution of chloroplast density, stomatal morphology, and the patterns of chloroplasts and stomata in cacao leaves during the flush growth Stage C. The study utilizes microscopic data to reveal variations in chloroplast and stomatal size and distribution across the leaf. A machine learning approach is employed to analyze these patterns, which could potentially be applied to a broader range of species and developmental stages. Overall, The paper presents a novel application of machine learning to understand the developmental biology of cacao leaves. It is well-structured, with clear claims and a logical progression of ideas.

Minor Comments:

1. The authors counted chloroplast clusters in the leaves, but it would be helpful to clarify how the size of these clusters is distinguished. Providing an example image of both larger and smaller clusters in Figure 1 would enhance understanding.
2. In Figure 3(b), the largest stomata are predominantly localized in the G1 area, which seems to differ significantly from the conclusions drawn. How can this discrepancy be explained?
3. In cacao leaves at Stage C, it appears that the differences in stomatal size may be related to their developmental stages, with older stomata being larger and younger stomata smaller. The authors should clarify whether the stomata are fully mature or if they are developing in different areas of the leaves.

Reviewer #2

(Remarks to the Author)

The manuscript by Baek et al. describes the spatial patterning of chloroplasts and stomata in the growth C stage leaves of Cacao, an important but less studied tropical tree species. The authors captured tons of morphological characteristics on the cacao leaf segments, chloroplast clusters, and stomata, following with intensive correlation and hierarchical analysis by machine-learning SVM-based classification. Finally, they summarized the strong correlations between chloroplast development and stomatal size. This study not only contributes to our fundamental knowledge on the cacao developmental biology, but also provides a novel machine-learning solution in analyzing large data for agricultural/plant researchers.

Major comments

1. Overall, the manuscript didn't read smoothly.
 - a. Logic in writing. Some text contents didn't match with sequence of figures. For example, it would be more helpful for readers without background knowledge in cacao biology to learn the features of cacao leaf developmental stages by moving Fig. 8. to Fig. 1.

b. Data presentation style(s). It is hard to read and interpret data from Fig. 2., Fig. 3., and Fig. 4. It would be much more straightforward to show the size(s) variations from different leaf segments by adding bot-plot or violet plot images showing details such as average, standard deviation, outliers, etc.

c. Grammar. The current manuscript needs to be polished by a native English speaker or a professional language editing service with plant biology/agricultural research background.

2. Unclear definitions of key parameters.

a. Definition and calculation of 'Chloroplast clusters'. Please add an image, and/or formula to explain it. Does it mean measurements on size of chloroplast-forming clusters in a single cell, or clusters from different cells? Please specify which cell type. Guard cells? Mesophyll cells? Or both guard cells and mesophyll cells? Plus, it is not possible to identify single chloroplast from Fig 1(b) because of the low resolution of the image. How to define the boundaries between chloroplast clusters and the background of images? Also, what does 'Localized groups of chloroplasts' in Fig. 1(b) mean? Venules-localized? However, from Fig 1(b), it is clear that venules are not green-labeled areas. Plus, what is the biological meaning/significance of measuring chloroplast clusters? Please provide more information.

b. Distribution pattern of stomata does not equal to size of stomata. If the authors wish to describe the distribution of stomata, it is necessary to get image data such as cell wall staining figure showing the distribution patterns of precursor cells, guard cells, pavement cells, and trichomes, as well as numerical data such as stomatal index, stomatal density, etc. Besides, I am not clear about how 'Stomatal circularity' is calculated and its biological meaning. Since the guard cell movement regulates the opening and closing of stomatal pore, even for the same stomatal complex, its circularity would be affected by change of environmental cues such as light, humidity, and CO₂ concentration, etc.

3. Concerns in the methods.

a. In Materials and Methods-Leaf scanning and microscopy. 'All images were saved in JPEG format for further analysis'. The most common used and acceptable digital image format in scientific research is TIFF. JPEG does not store much information as TIFF.

b. In Materials and Methods-Leaf scanning and microscopy, the authors mentioned 'Chloroplast clusters were analyzed at 10X, with clusters connected by more than 50% of their area considered a single unit', while in Image Analysis, 'However, chloroplast clusters and stomata required manual measurement, as automated detection was not possible'. How could the author try to avoid artificial bias when manually define the boundaries between chloroplasts and background of the images? Any filter/mask used to reduce the background noises? Also, from the Fig. 1(b), at 20X image, the hand drawing outlines of guard cells are not reflecting the kidney-shaped guard cells. Without clear plant cell wall labeling such as propidium iodide staining, the accuracy of the stomatal complex area would be affected significantly by this manual measurement method.

c. In Image Analysis, 'Leaf greenness was measuring RGB values (0-255 scale, 0=highest intensity)...and freehand selection was used to measure specific regions'. Please provide more details about how to process the raw images, such as background extraction, signal threshold setting, to separate the green area from the background with less artificial bias.

d. To describe the greenness of each leaf segment, I am not convinced by App-based image data only, such as the values of greenness and size of chloroplast clusters/circularity. It would be more convincible to get traditional wet bench data like chlorophyll concentration measurements and correlate both dry bench data and wet bench data.

e. In the paragraph after Fig.6, the authors mentioned '...trained on 80% of the data, while the remaining 20% was used for testing', it is not reasonable to use same dataset by SVM to test the accuracy of SVM training and testing groups without informing readers how the data was measured. By human? By SmartGrain? Or by SVM? It is quite confusing: in earlier main text, it seemed that they captured data via SmartGrain, however, in the method description, they said '...chloroplast clusters and stomata required manual measurement, as automated detection was not possible'. From my understanding, it is meaningful to compare human-measured data and App-measured data from the same sample, and calculate the correlation rates. Ref.13 is a good example.

Minor comments

1. No scale bar(s) in Fig.1(b).
2. Fig.5. When mentioning parameters for stomata, use the adjective word 'stomatal' in stomatal length, stomatal perimeter, etc.
3. NO.9 citation format is not consistent with other citations.
4. Materials and Methods-Plant materials: didn't mention the temperature of the greenhouse.
5. How many days in stage C? On which day did the authors get the samples?
6. Didn't mention bit-depth of original images.
7. No line numbers/page numbers. Hard to follow.
8. Please add a table to list leaf/chloroplast/stomatal traits and their abbreviation, description, and unit.

(Remarks to the Author)

I have thoroughly reviewed the manuscript "Spatial Patterning of Chloroplasts and Stomata in Developing Cacao Leaves." While research on the cacao plant is valuable and highly appreciated, I found that this paper is not well-prepared and lacks clarity in several areas. The overall presentation of the work does not sufficiently convey the objectives, methodology, and findings clearly and coherently. Further refinement in both the structure and the explanation of the research is needed to make the contribution more accessible and impactful to the readers.

1. The paper lacks focus and clarity, making it difficult to determine whether the authors report on 1) cacao stomatal phenotyping, 2) cacao stomatal development, or 3) cacao leaf physiology. The manuscript does not seem to provide enough concrete data to support any of these aspects adequately. The authors should consider restructuring the paper to ensure a more focused approach and clearly define the specific research questions they aim to address. I strongly encourage the authors to revise the manuscript and refine the overall direction of the study.

2. The data presentation is overly complex and somewhat confusing, making it challenging to interpret the results. The figures could benefit from simplified formatting and better integration into the narrative. I recommend revising the data presentation to enhance clarity and ensure readers can easily follow the key findings.

3. The manuscript does not provide clear evidence or data supporting the claim of hormonal regulation. Given the study's focus, I encourage the authors to incorporate experimental data on the role of hormones, mainly at various growth cycle stages. This would help strengthen the argument and provide a more comprehensive understanding of the physiological processes being studied. Including data on hormonal regulation would be crucial in validating the manuscript's claims.

4. It is unclear whether abscisic acid (ABA) is the primary focus of this paper. If ABA is central to the study, I recommend the authors design specific experiments to examine its role in cacao physiology at different stages of growth. Alternatively, suppose the paper is focused on stomatal development or other aspects of leaf physiology. In that case, the authors should streamline the content to emphasize those areas and remove unnecessary references to ABA. A clear research direction is needed to avoid confusion.

5. The manuscript does not specify which particular stomatal traits were measured using the Smart Grain software. It would be helpful if the authors could explicitly list the traits analyzed, such as stomatal density, index, or pore size, and clarify how these measurements contribute to the overall study. Providing this detail would improve the transparency of the methodology and allow readers to assess the reliability of the results better.

6. Earlier in the manuscript, the authors mentioned using SmartGrain to measure stomatal traits, yet later in the text, they assert that stomata measurements were done manually. This contradiction is confusing and undermines the clarity of the methodology section. I recommend the authors clarify whether SmartGrain was used for all measurements or whether manual measurements were involved. Consistency in the description of the methods is crucial for reproducibility.

7. While the Smart Grain software is mentioned in the manuscript as a tool for measuring stomatal traits, it is essential to note that such software may not be particularly innovative in the context of current research. There are numerous well-established and easily accessible resources and tools available for this type of analysis. For instance, studies like those in DOI: 10.1093/plphys/kiac049 and freely available AI-driven tools such as Biodock are widely used by researchers in the field. These resources can perform similar tasks with greater efficiency and accessibility, making the contribution of SmartGrain in this study seem less novel. I recommend that the authors consider how their methodology fits within the broader landscape of existing tools and possibly highlight any unique aspects that set this work apart from established alternatives.

8. The manuscript lacks significant references to key literature in the field. Specifically, studies exploring stomata's micro and macro heterogeneity were conducted long before this study, providing essential context for understanding the current research. A notable example is the work cited in DOI: 10.1016/S0065-2296(08)60124-X, which presents a comprehensive understanding of stomatal variability. The authors would benefit from reviewing this and other relevant studies to position their work within the existing body of knowledge and well-established findings. A more thorough literature review would strengthen the manuscript and demonstrate an awareness of previous research on this topic.

Version 1:

Reviewer comments:

Reviewer #1

(Remarks to the Author)

The authors have already addressed my comments, and I have no further feedback.

Reviewer #2

(Remarks to the Author)

The revised manuscript reads more smoothly. The authors have satisfactorily addressed most of my concerns.

Minor 1. The authors had replaced 'localized groups of chloroplasts' with 'chloroplast clusters in the interveinal regions' or 'chloroplast clusters' in the main text. However, in figure 1b (between line 144 and line 145), the authors still used 'Localized groups of chloroplasts'. Please double check.

Minor 2. It is understandable that the SmartGrain (published in 2012) is not designed for measuring stomatal

morphology/chloroplast traits so that the authors had to measure thousands of data points manually. However, I still need to emphasize my points 1) The purpose of machine learning-based apps/tools on stomatal morphology is to free our hands and accelerate the speed of measurements/analysis. 2) Dataset captured by machine learning-based apps/tools should be correlated to manually captured dataset to evaluate the accuracy of apps/tools. I wish the authors could use some state of the art machine learning-based tools in their future studies to do high-throughput data collection and correlate the results with human captured data, if possible.

Look forward to seeing more cacao developmental and physiological studies in the future. Good luck!

Reviewer #4

(Remarks to the Author)

Here, Baek et al. investigated the spatial relationship between chloroplast and stomatal development in cacao leaves during growth Stage C. The diagrams effectively illustrate the characters of chloroplast clusters and stomata across different interveinal segmentations from the leaf tip to the base. Additionally, the authors employ an SVM-RBF learning approach to reveal distinct patterns in the variation of chloroplasts and stomata within specific leaf regions. Overall, the revised manuscript is clearer and more accessible.

My comments and suggestions are as follows,

Major points:

1. While the study focuses on spatial patterns, future research could explore whether phytohormone signaling plays a role in establishing these patterns. This lies beyond the current scope but represents an intriguing direction for further investigation. Additionally, the spatial variation trends of chloroplast clusters and stomatal size appear wavelike, as shown in Fig. S2a and S4a. Could these patterns be linked to dynamic circadian growth processes?
2. Stomatal density and aperture are critical parameters that reflect stomatal characteristics during leaf development. Including these indicators in the analysis would strengthen the study's conclusions and provide a more comprehensive understanding of stomatal behavior.
3. The study highlights correlations between chlorophyll accumulation, stomatal morphology, and growth patterns. However, the causal relationships among these factors remain unclear.
4. While three biological replicates are relatively limited given the ontogenetic differences, the trends are consistent across most figures except for Fig. 3b. To ensure reliability, additional biological replicates for Fig. 3b would be beneficial.

Overall, this manuscript presents valuable insights into the spatial coordination of chloroplast and stomatal development during cacao leaf maturation. With the inclusion of additional experimental data to address the points above, I believe this work is suitable for publication.

Version 2:

Reviewer comments:

Reviewer #4

(Remarks to the Author)

I appreciate the authors' detailed explanation regarding the time-intensive nature of the additional experiments suggested. Given the substantial time investment required for manual microscopic analysis and the seasonal constraints of cacao leaf development, I fully understand the practical challenges in expanding the biological replicates at this stage. Therefore, I have no further concerns regarding the current manuscript.

Reviewers' comments:

Reviewer #1 (Remarks to the Author):

The article titled "Spatial Patterning of Chloroplasts and Stomata in Developing Cacao Leaves" by Insuck Baek et al. investigates the spatial distribution of chloroplast density, stomatal morphology, and the patterns of chloroplasts and stomata in cacao leaves during the flush growth Stage C. The study utilizes microscopic data to reveal variations in chloroplast and stomatal size and distribution across the leaf. A machine learning approach is employed to analyze these patterns, which could potentially be applied to a broader range of species and developmental stages. Overall, The paper presents a novel application of machine learning to understand the developmental biology of cacao leaves. It is well-structured, with clear claims and a logical progression of ideas.

R: Thank you so much for your comment.

Minor Comments:

1. The authors counted chloroplast clusters in the leaves, but it would be helpful to clarify how the size of these clusters is distinguished. Providing an example image of both larger and smaller clusters in Figure 1 would enhance understanding.

R: We acknowledge that further clarification about the size of chloroplast clusters would be helpful. In the manuscript, we have included a clearer statement in the Materials and Methods section, as well as example images of both larger and smaller clusters in Figure 1.

2. In Figure 3(b), the largest stomata are predominantly localized in the G1 area, which seems to differ significantly from the conclusions drawn. How can this discrepancy be explained?

The reviewer correctly noted that Leaf 2 in Figure 3b shows a unique pattern, with larger stomata in the G1 region, unlike the general acropetal trend seen in other leaves. This may indicate the dynamic nature of leaf development in Stage C, with Leaf 2 possibly representing an earlier phase. The influence of the developing midrib and primary veins on stomatal initiation might be more pronounced at this stage. Additionally, variations in the controlled greenhouse environment and developmental plasticity could contribute to this pattern. While there may be minor technical artifacts, the consistent chloroplast data and distinct stomatal size differences in Leaf 2 suggest a biological basis for the findings.

3. In cacao leaves at Stage C, it appears that the differences in stomatal size may be related to their developmental stages, with older stomata being larger and younger stomata smaller. The authors should clarify whether the stomata are fully mature or if they are developing in different areas of the leaves.

R: We thank the reviewer for their insightful question regarding stomatal maturity. We agree that size differences reflect developmental stages. In Stage C cacao leaves, which

are still developing, we interpret 'larger' stomata as being further along in their developmental trajectory towards full functionality compared to 'smaller' stomata, particularly those in apical regions. We added a sentence in the discussion to make it clearer. Thank you so much for the insightful review.

Reviewer #2 (Remarks to the Author):

The manuscript by Baek et al. describes the spatial patterning of chloroplasts and stomata in the growth C stage leaves of Cacao, an important but less studied tropical tree species. The authors captured tons of morphological characteristics on the cacao leaf segments, chloroplast clusters, and stomata, following with intensive correlation and hierarchical analysis by machine-learning SVM-based classification. Finally, they summarized the strong correlations between chloroplast development and stomatal size. This study not only contributes to our fundamental knowledge on the cacao developmental biology, but also provides a novel machine-learning solution in analyzing large data for agricultural/plant researchers.

Major comments

1. Overall, the manuscript didn't read smoothly.

R: We thank the reviewer for their feedback regarding the overall flow of the manuscript. We have carefully revised the entire manuscript to improve its readability and ensure a smoother, more logical progression of ideas.

a. Logic in writing. Some text contents didn't match with sequence of figures. For example, it would be more helpful for readers without background knowledge in cacao biology to learn the features of cacao leaf developmental stages by moving Fig. 8. to Fig. 1.

R: We appreciate the reviewer's suggestion regarding the logical flow and the placement of Figure 8. We understand the need to provide sufficient background information on cacao leaf developmental stages for readers unfamiliar with this species. However, we believe that placing Figure 8 as Figure 1 would be premature, as this figure presents a hypothetical model derived from the results presented in the current study. It is best understood after the data on chloroplast and stomatal distribution have been introduced. To address the reviewer's concern about background knowledge, we have reinforced the description of cacao leaf developmental stages in the Introduction section, providing more context on the characteristics of each stage (A, B, C, D, and E).

b. Data presentation style(s). It is hard to read and interpret data from Fig. 2., Fig. 3., and Fig. 4. It would be much more straightforward to show the size(s) variations from different leaf segments by adding bot-plot or violet plot images showing details such as average, standard deviation, outliers, etc.

R: We appreciate the reviewer's suggestion to use box plots or violin plots to visualize the data in Figures 2, 3, and 4. We understand that these plots can be effective in showing summary statistics like averages, standard deviations, and outliers.

While we believe the current heatmap format in these figures is crucial for highlighting the spatial aspect of our data, which is a central focus of our study, we also recognize the value of providing summary statistics. To address this, we have included supplementary figures (Fig. S2 and S4) that display the mean and 95% confidence intervals for the chloroplast and stomatal morphology traits, respectively.

c. Grammar. The current manuscript needs to be polished by a native English speaker or a professional language editing service with plant biology/agricultural research background.

R: "We appreciate the reviewer's feedback on the language. The manuscript has been reviewed by native English-speaking co-authors and revised accordingly to improve clarity and flow.

2. Unclear definitions of key parameters.

a. Definition and calculation of 'Chloroplast clusters'. Please add an image, and/or formula to explain it. Does it mean measurements on size of chloroplast-forming clusters in a single cell, or clusters from different cells?

R: We thank the reviewer for raising the need to clarify the term "chloroplast clusters." In our study, "chloroplast clusters" refer to groups of individual chloroplasts observed in close proximity within a defined area of the interveinal leaf tissue under 10X magnification. These clusters may originate from multiple cells and were not quantified on a per-cell basis. Clusters connected by more than 50% of their area were considered a single unit. We have revised the Methods section to provide a more explicit definition of chloroplast clusters and have updated the Figure 1 legend to specify that the images illustrate examples of these clusters as observed in our analysis.

Please specify which cell type. Guard cells? Mesophyll cells? Or both guard cells and mesophyll cells?

R: We thank the reviewer for their question regarding the cell types included in our analysis. Our measurements of stomatal morphology were, by definition, focused on guard cells, as stated in materials and methods.

Plus, it is not possible to identify single chloroplast from Fig 1(b) because of the low resolution of the image. How to define the boundaries between chloroplast clusters and the background of images?

R: We thank the reviewer for their observation regarding the resolution of the images and the challenge of identifying individual chloroplasts. We acknowledge that at the magnification used for analyzing chloroplast clusters (10X), it is difficult to discern individual chloroplasts clearly. Our analysis focused on chloroplast clusters, which we defined as groups of chloroplasts in close proximity within the interveinal areas, rather than attempting to quantify individual chloroplasts. We apologize for causing any confusion.

Also, what does 'Localized groups of chloroplasts' in Fig. 1(b) mean? Venules-localized? However, from Fig 1(b), it is clear that venules are not green-labeled areas.

R: We thank the reviewer for their careful observation regarding the terminology used in Figure 1(b). We apologize for any confusion caused by the phrase 'localized groups of chloroplasts.' We used this term to refer to chloroplast clusters found within the interveinal regions of the leaf, as seen in the 10X magnification images. The reviewer is correct that the venules themselves are not the primary location of these clusters. The green-labeled areas in Figure 1(b) represent areas with higher concentrations of chloroplasts within the mesophyll cells. We have revised the Figure 1 legend and other relevant sections of the manuscript to clarify this distinction. Specifically, we replaced 'localized groups of chloroplasts' with 'chloroplast clusters in the interveinal regions' or simply 'chloroplast clusters' where appropriate to avoid ambiguity.

Plus, what is the biological meaning/significance of measuring chloroplast clusters? Please provide more information.

R: We thank the reviewer for prompting us to elaborate on the biological significance of measuring chloroplast clusters. While individual chloroplast morphology and function are undoubtedly important, we chose to analyze chloroplast clusters in this study because they represent a higher level of organization within the leaf tissue that is likely relevant to overall photosynthetic capacity and leaf development. Chloroplast distribution within leaves is not uniform, and analyzing clusters allows us to capture this spatial heterogeneity and relate it to other developmental gradients, such as stomatal size and leaf greenness. Furthermore, the size, shape, and distribution of chloroplast clusters may reflect the developmental stage of the mesophyll tissue and its progression towards full photosynthetic competence. The proximity of chloroplasts within a cluster could potentially influence light capture, energy transfer, and overall photosynthetic efficiency, and while we did not directly measure these functional parameters in this study, chloroplast cluster morphology could serve as a useful proxy for these aspects. Given the limited research on cacao leaf development, particularly at the tissue and organ level, analyzing chloroplast clusters provides a novel perspective on this important process in a valuable crop species. We have expanded the Discussion section to elaborate on these points and highlight the potential implications of our findings for understanding cacao leaf development and photosynthetic efficiency.

b. Distribution pattern of stomata does not equal to size of stomata. If the authors wish to describe the distribution of stomata, it is necessary to get image data such as cell wall staining figure showing the distribution patterns of precursor cells, guard cells, pavement cells, and

trichomes, as well as numerical data such as stomatal index, stomatal density, etc. Besides, I am not clear about how 'Stomatal circularity' is calculated and its biological meaning. Since the guard cell movement regulates the opening and closing of stomatal pore, even for the same stomatal complex, its circularity would be affected by change of environmental cues such as light, humidity, and CO₂ concentration, etc.

R: We thank the reviewer for their insightful comments on stomatal distribution and size, and the clarification regarding stomatal circularity.

We acknowledge the difference between stomatal distribution and size. In this study, we focused on spatial variation in stomatal size as an indicator of developmental stage, rather than on distribution patterns. To clarify, we now consistently use the term "spatial variation in stomatal size" in the manuscript.

The reviewer is correct that environmental factors influence stomatal aperture and circularity. However, our measurements were taken from fixed leaf samples under consistent conditions. We used circularity as a morphometric descriptor of stomatal shape, which varied across the leaf. We hypothesized that these variations reflect different developmental stages of the stomata. We have added details on how circularity was calculated in the Methods section and discussed the biological implications of these variations.

3. Concerns in the methods.

a. In Materials and Methods-Leaf scanning and microscopy. 'All images were saved in JPEG format for further analysis'. The most common used and acceptable digital image format in scientific research is TIFF. JPEG does not store much information as TIFF.

R: We appreciate the reviewer highlighting the issue of image format. We recognize that TIFF is typically the preferred format for scientific imaging because it employs lossless compression. In our future studies, we will pay closer attention to the choice of image formats.

b. In Materials and Methods-Leaf scanning and microscopy, the authors mentioned 'Chloroplast clusters were analyzed at 10X, with clusters connected by more than 50% of their area considered a single unit', while in Image Analysis, 'However, chloroplast clusters and stomata required manual measurement, as automated detection was not possible'. How could the author try to avoid artificial bias when manually define the boundaries between chloroplasts and background of the images? Any filter/mask used to reduce the background noises? Also, from the Fig. 1(b), at 20X image, the hand drawing outlines of guard cells are not reflecting the kidney-shaped guard cells. Without clear plant cell wall labeling such as propidium iodide staining, the accuracy of the stomatal complex area would be affected significantly by this manual measurement method.

R: We thank the reviewer for their careful consideration of our methods and for raising the important questions of potential bias in manual image analysis.

We recognize that manual segmentation of chloroplast clusters can be subjective. To minimize bias, we established clear criteria for cluster definition, focusing on groups of chloroplasts in close proximity within interveinal regions, with clusters connected by over 50% of their area considered a single unit. Measurements were conducted blind to further reduce bias. While we did not apply additional filters beyond the SmartGrain software's auto enhancement, we acknowledge this as a potential area for improvement in future studies.

Regarding the outlines of guard cells in Figure 1(b): We acknowledge that the hand-drawn outlines in Figure 1(b) are **simplified representations for illustrative purposes** and may not perfectly capture the kidney-shaped morphology of the guard cells in the cacao genotype used. The actual measurements of stomatal size were performed using the SmartGrain software, which allows for more precise tracing of the guard cell boundaries. Variations in guard cell morphology can occur between plant species, genotypes, and even due to the angle of imaging. Our team, with expertise in plant biology (a few members), made every effort to accurately trace the guard cell boundaries based on established morphological criteria.

We agree with the reviewer that using cell wall stains like propidium iodide could enhance the accuracy of cell boundary identification. However, this was not feasible in our current study, as it would have precluded the analysis of chloroplast clusters and stomatal size in the same leaf samples. We currently have other projects with deep learning applications to analyze stomata involved in staining whole leaves, which clearly shows stomata morphology. Compared to the stained images, we think the analysis we performed here shows a similar overall stomata shape. We aimed to investigate the relationship between these two features, necessitating a non-destructive imaging approach.

c. In Image Analysis, 'Leaf greenness was measuring RGB values (0-255 scale, 0=highest intensity)...and freehand selection was used to measure specific regions'. Please provide more details about how to process the raw images, such as background extraction, signal threshold setting, to separate the green area from the background with less artificial bias.

R: We thank the reviewer for their question regarding the quantification of leaf greenness. We used ImageJ to measure the mean gray value of the green channel within manually selected leaf areas in the images. The 'Split Channels' function was used to separate the RGB channels, and the freehand tool was used to carefully outline the leaf or leaf segment, excluding the background. We did not use automated background extraction or thresholding, as the uniform background allowed for clear visual discrimination of leaf edges. We have expanded the Methods section to provide a more detailed description of this process.

d. To describe the greenness of each leaf segment, I am not convinced by App-based image data only, such as the values of greenness and size of chloroplast clusters/circularity. It would be more convincing to get traditional wet bench data like chlorophyll concentration measurements and correlate both dry bench data and wet bench data.

R: We thank the reviewer for their suggestion regarding the inclusion of traditional chlorophyll concentration measurements. We agree that wet-bench data, such as chlorophyll extraction and quantification, provide valuable and widely accepted measures of leaf greenness.

However, the primary focus of this initial study was to explore the spatial variation in leaf greenness and its relationship to chloroplast cluster and stomatal morphology across the developing leaf, using image-based analysis as a novel and relatively high-throughput approach. Our image-based method allowed us to quantify greenness in multiple segments of the same leaf, revealing spatial patterns that would be difficult to capture using traditional bulk measurements.

While we did not include chlorophyll extraction data in this study, we acknowledge its importance and have plans to incorporate such measurements in future research. This will involve an extensive collection of cacao genotypes and a more in-depth analysis of chlorophyll content, including different chlorophyll types, and their correlation with image-based data, as well as various physiological parameters.

We believe that our current study provides valuable preliminary insights into the spatial dynamics of leaf development in cacao and lays the groundwork for these future investigations.

e. In the paragraph after Fig.6, the authors mentioned ‘...trained on 80% of the data, while the remaining 20% was used for testing’, it is not reasonable to use same dataset by SVM to test the accuracy of SVM training and testing groups without informing readers how the data was measured. By human? By SmartGrain? Or by SVM? It is quite confusing: in earlier main text, it seemed that they captured data via SmartGrain, however, in the method description, they said ‘...chloroplast clusters and stomata required manual measurement, as automated detection was not possible’. From my understanding, it is meaningful to compare human-measured data and App-measured data from the same sample, and calculate the correlation rates. Ref.13 is a good example.

R: We thank the reviewer for their comment regarding the SVM analysis and the method of data measurement. We apologize for any confusion regarding this aspect of our methodology. We would like to clarify that the data used for SVM training and testing were obtained through manual measurements with the assistance of the SmartGrain software, as described in the Methods section.

To reiterate, the SmartGrain software was used as a tool to aid in the manual tracing and quantification of chloroplast clusters and stomata from microscopic images. The software's edge detection algorithms helped define the boundaries of these features based on color intensity differences, but the final delineation and measurement were performed manually by a trained researcher. This is what we meant by 'manual measurement, as automated detection was not possible' in the Methods section.

The 80/20 split for SVM training and testing refers to dividing this manually obtained dataset into two subsets: 80% of the samples were used to train the SVM model, and the

remaining 20% were used to test its performance on unseen data. This is a standard practice in machine learning to evaluate the model's ability to generalize to new data.

We agree with the reviewer that comparing manual measurements with automated measurements could be valuable. However, in our case, SmartGrain did not perform automated measurements; it assisted with manual ones. Therefore, a direct comparison in the manner suggested by the reviewer is not applicable in this context.

We have revised the Methods section to further clarify the role of SmartGrain in our analysis and the process of manual measurement.

Minor comments

1. No scale bar(s) in Fig.1(b). **R: Figure 1(b) was primarily intended as a representative illustration of the different leaf segments and the general appearance of chloroplast clusters and stomata under the microscope. While the newly added insets highlight examples of chloroplast cluster size differences, the primary purpose of the main panel (b) remains illustrative rather than quantitative.**
2. Fig.5. When mentioning parameters for stomata, use the adjective word 'stomatal' in stomatal length, stomatal perimeter, etc. **R: We thank the reviewer for their suggestion to use the adjective "stomatal" when referring to stomatal parameters. We agree that this improves clarity and have made the necessary changes throughout the manuscript.**
3. NO.9 citation format is not consistent with other citations. **R: We revised the manuscript as suggested.**
4. Materials and Methods-Plant materials: didn't mention the temperature of the greenhouse. **R: We added more detailed info regarding greenhouse conditions.**
5. How many days in stage C? On which day did the authors get the samples? **R: We thank the reviewer for their questions regarding the timing of sampling within Stage C. Our samples were collected from leaves visually assessed to be in Stage C based on their characteristic coloration (transitioning from brown to light green) and active expansion, as described in the Introduction and Methods. While we did not record the exact day of sampling within Stage C, all leaves were sampled during a similar developmental window based on these visual criteria.**
6. Didn't mention bit-depth of original images. **R: We thank the reviewer for raising the question about the bit depth of our images. The original images used for analysis were captured in 24-bit RGB color. We have added this information to the Methods section of the manuscript under 'Leaf scanning and microscopy'.**

7. No line numbers/page numbers. Hard to follow.

R: We thank the reviewer for their feedback regarding the lack of line and page numbers. We inserted line number/page numbers to address this issue.

8. Please add a table to list leaf/chloroplast/stomatal traits and their abbreviation, description, and unit.

R: We thank the reviewer for the suggestion to include a table summarizing the traits and their abbreviations. We agree that this will be helpful for readers. We have created a new supplementary table (Table S1) that lists all leaf, chloroplast, and stomatal traits measured in this study, along with their abbreviations, descriptions, and units.

Reviewer #3 (Remarks to the Author):

I have thoroughly reviewed the manuscript "Spatial Patterning of Chloroplasts and Stomata in Developing Cacao Leaves." While research on the cacao plant is valuable and highly appreciated, I found that this paper is not well-prepared and lacks clarity in several areas. The overall presentation of the work does not sufficiently convey the objectives, methodology, and findings clearly and coherently. Further refinement in both the structure and the explanation of the research is needed to make the contribution more accessible and impactful to the readers.

1. The paper lacks focus and clarity, making it difficult to determine whether the authors report on 1) cacao stomatal phenotyping, 2) cacao stomatal development, or 3) cacao leaf physiology. The manuscript does not seem to provide enough concrete data to support any of these aspects adequately. The authors should consider restructuring the paper to ensure a more focused approach and clearly define the specific research questions they aim to address. I strongly encourage the authors to revise the manuscript and refine the overall direction of the study.

R: We sincerely thank the reviewer for their insightful feedback and for highlighting the need to clarify the focus of our study. We understand their concern that the manuscript lacked a clear direction and may have appeared to cover multiple aspects without sufficient depth.

2. The data presentation is overly complex and somewhat confusing, making it challenging to interpret the results. The figures could benefit from simplified formatting and better integration into the narrative. I recommend revising the data presentation to enhance clarity and ensure readers can easily follow the key findings.

R: We thank the reviewer for their feedback regarding the complexity of our data presentation. We understand that the detailed nature of our figures, particularly the heatmaps, might initially appear challenging to interpret. However, these figures were designed to illustrate the spatial variations in stomatal and chloroplast characteristics, which is a central aspect of our study. Simplifying these figures could result in a loss of

valuable spatial information. **To address the reviewer's concern, we have included supplementary figures (Fig. S2 and S4) that display the mean and 95% confidence intervals for the chloroplast and stomatal morphology traits, respectively.**

To enhance clarity and address the reviewer's concerns, we have taken the following steps- To improve clarity, we have revised the legends of Figures, clarified terminology regarding chloroplast clusters and stomatal size, added a new Supplementary Table (Table S1) listing all measured traits with their definitions, and incorporated introductory sentences to better guide the reader through the figures.

3. The manuscript does not provide clear evidence or data supporting the claim of hormonal regulation. Given the study's focus, I encourage the authors to incorporate experimental data on the role of hormones, mainly at various growth cycle stages. This would help strengthen the argument and provide a more comprehensive understanding of the physiological processes being studied. Including data on hormonal regulation would be crucial in validating the manuscript's claims.

We thank the reviewer for their comment regarding the role of hormonal regulation. We agree that hormonal regulation likely plays a significant role in cacao leaf development, including the processes we describe. However, the mention of hormonal regulation, specifically abscisic acid, in the Discussion was intended to be hypothetical and to suggest potential avenues for future research, rather than a central claim of the current study. Our primary focus in this manuscript was to characterize the spatial variations in chloroplast and stomatal morphology during leaf development and to explore their relationship with leaf greenness, not to provide a comprehensive analysis of hormonal regulation.

We acknowledge that we did not present experimental data on hormone levels or activity in this study. This was because our initial aim was to establish a solid understanding of the spatial and temporal dynamics of leaf development through these detailed morphological observations. We strongly agree with the reviewer that investigating the role of hormones, especially ABA, is crucial for a more comprehensive understanding of these processes. Indeed, we see this as a critical next step, and we are planning future studies that will incorporate such analyses, including hormone quantification and manipulation experiments at different stages of the growth cycle.

4. It is unclear whether abscisic acid (ABA) is the primary focus of this paper. If ABA is central to the study, I recommend the authors design specific experiments to examine its role in cacao physiology at different stages of growth. Alternatively, suppose the paper is focused on stomatal development or other aspects of leaf physiology. In that case, the authors should streamline the content to emphasize those areas and remove unnecessary references to ABA. A clear research direction is needed to avoid confusion.

R: We appreciate the reviewer's insightful comment about ABA. While ABA is not the main focus of our paper, we briefly mentioned its potential influence on chloroplast and stomatal morphology in relation to leaf greenness as part of the broader physiological context. We clarified this in the revised Discussion to reduce emphasis on ABA, ensuring our primary findings remain clear while acknowledging the interplay of various factors influencing leaf development.

5. The manuscript does not specify which particular stomatal traits were measured using the Smart Grain software. It would be helpful if the authors could explicitly list the traits analyzed, such as stomatal density, index, or pore size, and clarify how these measurements contribute to the overall study. Providing this detail would improve the transparency of the methodology and allow readers to assess the reliability of the results better.

We appreciate the reviewer's request for clarification regarding the stomatal traits measured with SmartGrain. We apologize for any confusion in our initial description. We measured stomatal area, perimeter, length, width, length-width ratio, circularity, and the distance from the intersection of length and width to the stomatal center of gravity. These traits are detailed in Supplementary Table S1, which lists all measured traits, their abbreviations, descriptions, and units. Although we did not measure stomatal density or index, we focused on these traits to explore variability during leaf development and their correlation with chloroplast clusters and leaf greenness. We have updated the Methods section to clearly outline the measured traits and reiterate the importance of Supplementary Table S1, enhancing the transparency of our methodology and addressing the reviewer's concerns.

6. Earlier in the manuscript, the authors mentioned using SmartGrain to measure stomatal traits, yet later in the text, they assert that stomata measurements were done manually. This contradiction is confusing and undermines the clarity of the methodology section. I recommend the authors clarify whether SmartGrain was used for all measurements or whether manual measurements were involved. Consistency in the description of the methods is crucial for reproducibility.

R: We sincerely apologize for the confusion regarding our description of the stomatal measurements. The reviewer is correct in pointing out the seeming contradiction between our earlier mention of using SmartGrain and our later statement about manual measurements. To clarify, we used the SmartGrain software as a tool to assist in the manual measurement of stomatal traits. While SmartGrain offers functionalities for automated measurements, these were not applicable to our specific images and the traits we needed to quantify. Therefore, we used SmartGrain's manual measurement capabilities. We understand that our initial description was a bit unclear and have revised the Methods section to explicitly state that SmartGrain was used to assist in, but not automate, the manual measurement of stomatal traits.

7. While the Smart Grain software is mentioned in the manuscript as a tool for measuring stomatal traits, it is essential to note that such software may not be particularly innovative in the context of current research. There are numerous well-established and easily accessible resources and tools available for this type of analysis. For instance, studies like those in DOI:

10.1093/plphys/kiae049 and freely available AI-driven tools such as Biodock are widely used by researchers in the field. These resources can perform similar tasks with greater efficiency and accessibility, making the contribution of SmartGrain in this study seem less novel. I recommend that the authors consider how their methodology fits within the broader landscape of existing tools and possibly highlight any unique aspects that set this work apart from established alternatives.

R: We thank the reviewer for suggesting alternative image analysis tools, including the example (DOI: 10.1093/plphys/kiae049) and AI platforms like Biodock. We appreciate the reviewer bringing these valuable resources to our attention.

We chose SmartGrain for this study due to its simplicity, compatibility with our images, and ability to measure specific traits relevant to our research (area, length, width, LWR, perimeter, circularity, and the distance between IS and CG). We were **unaware of more advanced tools when we began this study. While not the most cutting-edge, SmartGrain efficiently aided our manual measurement of numerous images.**

We agree that exploring advanced tools, including AI, could enhance future work. We are particularly interested in AI's potential for automating segmentation and measurement.

As stated in the Discussion, this study is a preliminary step in understanding spatial variations in chloroplast and stomatal morphology during cacao leaf development. Future studies will include more cacao genotypes and advanced image analysis. We will certainly explore the tools suggested by the reviewer.

8. The manuscript lacks significant references to key literature in the field. Specifically, studies exploring stomata's micro and macro heterogeneity were conducted long before this study, providing essential context for understanding the current research. A notable example is the work cited in DOI: 10.1016/S0065-2296(08)60124-X, which presents a comprehensive understanding of stomatal variability. The authors would benefit from reviewing this and other relevant studies to position their work within the existing body of knowledge and well-established findings. A more thorough literature review would strengthen the manuscript and demonstrate an awareness of previous research on this topic.

R: We thank the reviewer for highlighting the need for a more thorough literature review, particularly regarding stomatal heterogeneity. We have carefully reviewed the suggested reference (DOI: 10.1016/S0065-2296(08)60124-X) and other key studies on stomatal micro- and macro-heterogeneity. We have incorporated these into the Introduction and Discussion to provide a stronger context for our work. These additions demonstrate the relevance of our findings to the existing body of knowledge on stomatal variability.

Editor's comments:

Your manuscript entitled "Spatial patterning of chloroplasts and stomata in developing cacao leaves" has now been seen again by 3 referees, and I apologize for the delay in the processing of your manuscript. Note that Reviewer #3 was unable to look at the revised manuscript, so a new reviewer, Reviewer #4, was recruited as a replacement. You will see from the referees' comments below that while they find your work of considerable interest, some important points are raised. We are interested in the possibility of publishing your study in Communications Biology, but would like to consider your response to these concerns in the form of a revised manuscript before we make a final decision on publication.

A: Thank you for your consideration of our manuscript, and for managing the review process, including the recruitment of Reviewer #4. We appreciate the referees' positive feedback and recognize the importance of the points raised.

We therefore invite you to revise and resubmit your manuscript, taking into account the remaining concerns of Reviewer #2 as well as the points raised by Reviewer #4. In particular, we would strongly recommend adding additional biological replicates for Figure 3b as requested by Reviewer #4 (in their fourth comment).

A: Regarding Reviewer #4's fourth comment on adding additional biological replicates for Figure 3b, we sincerely appreciate the suggestion and recognize its potential to strengthen our analysis. However, as detailed in our response (see attached response to Reviewer #4, comment 4), the time-intensive nature of manual microscopic analysis, combined with the fixed flush growth cycle of cacao, made it infeasible to collect additional Stage C leaves within the current study's timeline. We have outlined our commitment to address this limitation in future research by developing a machine learning-based model for automated, high-throughput analysis, enabling larger sample sizes across multiple developmental stages. We believe this approach will resolve the concern and enhance future submissions to Communications Biology.

Please highlight all changes in the manuscript text file.

A: We have attached a highlighted version and a clean version of the revised manuscript to illustrate all changes made in response to the reviewers' comments.

We are committed to providing a fair and constructive peer-review process. Do not hesitate to contact us if you wish to discuss the revision in more detail or if there are specific requests from the reviewers that you believe are technically impossible or unlikely to yield a meaningful outcome.

A: Thank you so much!

At the same time, we ask that you ensure your manuscript complies with our editorial policies. Specifically:

For all graphs depicting a single point value (e.g., mean) with error bars, you must add individual data points or convert the graph to a boxplot or dot-plot to show data distribution.

A: This may apply to our Figures S2 and S4, which originally displayed graphs with means and 95% confidence intervals. To provide both a detailed view of data distribution and an overview, we have added boxplots alongside the original plots, allowing readers to access both comprehensive distributional information and a simplified summary simultaneously.

It's mandatory to provide access to the numerical source data for graphs and charts either through a repository or by providing the data in a Supplementary Data file (in excel format).

A: We have provided the raw numerical source data for all graphs and charts in a Supplementary Data file in Excel format.

All blots/gels must be accompanied by size markers in every figure panel. Uncropped and unedited blot/gel images must be included as Supplementary Figure(s) in the Supplementary Information pdf.

A: OK

Please ensure that you have complied with the data deposition policies at the Nature Portfolio, please see here.

A: OK

Please ensure that you have complied with our policies on research involving animals and humans, see here

A: OK

Please follow the ARRIVE guidelines for reporting animal experiments. Please fully complete an ARRIVE checklist including both the essential and recommended set of items (adding information to the manuscript where needed) and upload this with your revised manuscript.

A: OK

Please also see our revision checklist for guidance on formatting the manuscript and complying with our policies. A comprehensive guide to our formatting requirements for final submissions is also available for your reference here.

A: OK

Reviewers' comments:

Reviewer #1 (Remarks to the Author):

The authors have already addressed my comments, and I have no further feedback.

A: We thank the reviewer for their time and confirmation that their previous concerns have been satisfactorily addressed.

Reviewer #2 (Remarks to the Author):

The revised manuscript reads more smoothly. The authors have satisfactorily addressed most of my concerns.

A: We appreciate the reviewer's positive feedback and confirmation that their concerns have been largely addressed.

Minor 1. The authors had replaced 'localized groups of chloroplasts' with 'chloroplast clusters in the interveinal regions' or 'chloroplast clusters' in the main text. However, in figure 1b (between line 144 and line 145), the authors still used 'Localized groups of chloroplasts'. Please double check.

A: We appreciate the reviewer catching this inconsistency. The phrase 'Localized groups of chloroplasts' in Figure 1b has been replaced with 'chloroplast clusters', as suggested.

Minor 2. It is understandable that the SmartGrain (published in 2012) is not designed for measuring stomatal morphology/chloroplast traits so that the authors had to measure thousands of data points manually. However, I still need to emphasize my points 1) The purpose of machine learning-based apps/tools on stomatal morphology is to free our hands and accelerate the speed of measurements/analysis. 2) Dataset captured by machine learning-based apps/tools should be correlated to manually captured dataset to evaluate the accuracy of apps/tools. I wish the authors could use some state of the art machine learning-based tools in their future studies to do high-throughput data collection and correlate the results with human captured data, if possible.

A: We sincerely appreciate the reviewer's thoughtful feedback and suggestions about incorporating machine learning tools for analyzing stomata and chloroplasts. We agree that the primary aim of these tools is to boost efficiency and speed up data collection, and we recognize the critical importance of validating their results against manual measurements to ensure accuracy. In this study, we relied on manual measurements, but

we are actively planning to develop a machine learning-based model specifically designed for high-throughput analysis of stomatal and chloroplast cluster size and morphology in our future research.

Look forward to seeing more cacao developmental and physiological studies in the future. Good luck!

A: We thank the reviewer for their encouraging words and support for future research on cacao development and physiology.

Reviewer #3 (Remarks to the Author):-

A: NA, but we appreciate the first round of reviewer's sharp comments. Thank you.

Reviewer #4 (Remarks to the Author):

Here, Baek et al. investigated the spatial relationship between chloroplast and stomatal development in cacao leaves during growth Stage C. The diagrams effectively illustrate the characters of chloroplast clusters and stomata across different interveinal segmentations from the leaf tip to the base. Additionally, the authors employ an SVM-RBF learning approach to reveal distinct patterns in the variation of chloroplasts and stomata within specific leaf regions. Overall, the revised manuscript is clearer and more accessible.

My comments and suggestions are as follows,

Major points:

1. While the study focuses on spatial patterns, future research could explore whether phytohormone signaling plays a role in establishing these patterns. This lies beyond the current scope but represents an intriguing direction for further investigation. Additionally, the spatial variation trends of chloroplast clusters and stomatal size appear wavelike, as shown in Fig. S2a and S4a. Could these patterns be linked to dynamic circadian growth processes?

A: We thank the reviewer for the suggestions. We agree that exploring phytohormone signaling's role in shaping the spatial patterns of chloroplasts and stomata would deepen our understanding of flush growth mechanisms in cacao. However, as the reviewer notes, this exceeds the current study's scope, which focused on morphological and spatial analyses using microscopy and machine learning during Stage C. We have added a statement to the Discussion acknowledging this opportunity and proposing phytohormone studies for future investigation.

On the wavelike spatial variation trends in chloroplast clusters and stomatal size, we value the reviewer's observation. These patterns may indeed reflect dynamic circadian growth processes regulating diurnal rhythms in plant development. However, our study, designed to assess spatial heterogeneity at Stage C under controlled conditions, did not include time-series data or circadian analyses. We have included sentences in the Discussion recognizing this possibility and suggesting future research with time-resolved imaging and gene expression analyses to investigate circadian influences. These revisions address the limitations of our current approach and align with the reviewer's recommendations.

2. Stomatal density and aperture are critical parameters that reflect stomatal characteristics during leaf development. Including these indicators in the analysis would strengthen the study's conclusions and provide a more comprehensive understanding of stomatal behavior.

A: We appreciate the reviewer's valuable suggestion to include stomatal density and aperture in our analysis, as these parameters are indeed critical for a comprehensive understanding of stomatal behavior during cacao leaf development. However, in this study, we focused on morphological traits, which were manually measured using SmartGrain software due to the limitations of automated tools for accurately capturing these specific features in cacao leaves at Stage C. Measuring stomatal density and aperture would have required additional imaging techniques and extensive manual validation, which were beyond the scope of this study's resources and timeline. Nevertheless, we recognize the importance of these parameters and are committed to incorporating stomatal density and aperture measurements in our future research. We plan to develop and validate a machine learning-based model to enable automated, high-throughput analysis of these traits, including density across multiple developmental stages and cacao genotypes, thereby providing a more holistic understanding of stomatal behavior and its coordination with chloroplast development during flush growth.

3. The study highlights correlations between chlorophyll accumulation, stomatal morphology, and growth patterns. However, the causal relationships among these factors remain unclear.

A: We thank the reviewer for this insightful observation. We agree that while our study identifies robust correlations between chlorophyll accumulation, stomatal morphology, and spatial growth patterns in Stage C cacao leaves, it does not establish causality among these factors. This limitation is inherent to the observational and correlative nature of our current analysis, which focuses on describing spatial and morphological patterns during a specific developmental stage. To address this concern, we have added a statement to the Discussion section of the manuscript acknowledging the lack of causal evidence and suggesting future experimental approaches, such as genetic manipulation or time-series analyses of gene expression and physiological responses, to explore these relationships. We believe this addition clarifies the scope of our findings

and provides a roadmap for advancing our understanding of the mechanistic links between chlorophyll accumulation, stomatal development, and growth in cacao leaves.

4. While three biological replicates are relatively limited given the ontogenetic differences, the trends are consistent across most figures except for Fig. 3b. To ensure reliability, additional biological replicates for Fig. 3b would be beneficial.

A: We sincerely appreciate the reviewer's valuable feedback regarding the number of biological replicates used for the analysis presented in Figure 3b. We recognize that a larger sample size would enhance the statistical robustness of our findings, and we apologize that we were unable to include more replicates in the current study.

When we initiated this research, we underestimated the time-intensive nature of the manual microscopic analysis required for each leaf. The intricate process of segmenting leaves, imaging, and manually tracing thousands of chloroplast clusters and stomata proved to be significantly more time-consuming than initially anticipated. By the time we fully realized the extent of this limitation, we had already completed the analysis of the three Stage C leaves available to us, and the season had transitioned, making it impossible to obtain additional leaves at the comparable developmental stage until the next flush cycle.

Given the unique characteristics of cacao as a perennial crop with a fixed flush growth cycle, collecting additional leaves at the precise Stage C developmental stage required for this study is not currently feasible. Obtaining suitable samples would necessitate waiting for the next flush cycle, delaying the publication of these findings by at least six months, and repeating the entire, extensive microscopic analysis.

Our initial expectation, based on prior research and the inherent variability in plant development, was that cacao leaves would exhibit considerable heterogeneity. Therefore, the primary goal of this study was to establish a broad, foundational understanding of the spatial patterns of chloroplasts and stomata during this critical developmental phase, serving as a necessary first step for future, more in-depth investigations.

We are fully committed to addressing the issue of sample size and variability in our future research. We are actively developing a machine learning based model capable of automated and accurate measurement of stomatal and chloroplast characteristics (see reviewer 2 Minor comment #2). This will enable us to analyze a significantly larger number of samples, encompassing multiple developmental stages and cacao genotypes, and utilizing higher-resolution microscopy. This future work, facilitated by the automated analysis, will allow us to account for the inherent biological variability in cacao and the influence of environmental and seasonal factors, which can significantly impact leaf morphology. Without such automated methods, analyzing a substantially larger number of cacao leaf samples, given the time-sensitive nature of the flush growth and the

complexity of the manual measurements, is not practical within the scope of the current study. We believe that this planned, more comprehensive study, with increased sample sizes and automated analysis, will be a suitable candidate for future submission to Communications Biology.

Overall, this manuscript presents valuable insights into the spatial coordination of chloroplast and stomatal development during cacao leaf maturation. With the inclusion of additional experimental data to address the points above, I believe this work is suitable for publication.

A: We sincerely appreciate your positive and constructive feedback on our manuscript. We believe these revisions strengthen the manuscript and align with your suggestion for additional experimental data to enhance its impact. Thank you again for your thoughtful review and endorsement.